# Combining Ability and Gene Action Controlling Agronomic Traits for Cytoplasmic Male Sterile Line, Restorer Lines, and New Hybrids for Developing of New Drought-Tolerant Rice Hybrids

**DOI:** 10.3390/genes13050906

**Published:** 2022-05-19

**Authors:** Mamdouh M. A. Awad-Allah, Kotb A. Attia, Ahmad Alsayed Omar, Azza H. Mohamed, Rehab M. Habiba, Fahad Mohammed Alzuaibr, Mohammed Ali Alshehri, Mohammed Alqurashi, Salman Aloufi, Eldessoky S. Dessoky, Mohamed A. Abdein

**Affiliations:** 1Rice Research Department, Field Crops Research Institute, Agricultural Research Center, Giza 12619, Egypt; kattia1.c@ksu.edu.sa; 2Department of Biochemistry, College of Science, King Saud University, P.O. Box 2455, Riyadh 11451, Saudi Arabia; 3Biochemistry Department, Faculty of Agriculture, Zagazig University, Zagazig 44519, Egypt; aaelhanafi@agri.zu.edu.eg; 4Institute of Food and Agricultural Sciences, Citrus Research & Education Center, University of Florida, 700 Experiment Station Road, Lake Alfred, FL 33850, USA; azza@ufl.edu; 5Department of Agricultural Chemistry, College of Agriculture, Mansoura University, Mansoura 35516, Egypt; 6Department of Genetics, Faculty of Agriculture, Mansoura University, Mansoura, 35516, Egypt; rehab74@mans.edu.eg; 7Department of Biology, Faculty of Science, University of Tabuk, Tabuk 47713, Saudi Arabia; falzuaiber@ut.edu.sa (F.M.A.); ma.alshehri@ut.edu.sa (M.A.A.); 8Department of Biotechnology, Faculty of Science, Taif University, Taif 21974, Saudi Arabia; m.khader@tu.edu.sa (M.A.); s.aloufi@tu.edu.sa (S.A.); 9Department of Biology, College of Science, Taif University, P.O. Box 11099, Taif 21944, Saudi Arabia; es.dessouky@tu.edu.sa; 10Biology Department, Faculty of Arts and Science, Northern Border University, Rafha 91911, Saudi Arabia

**Keywords:** line × tester mating design, combining ability, drought management, *Oryza sativa* L., grain yield

## Abstract

This study aimed to identify new rice lines and hybrids that are tolerant to water deficit and produce high yields under water stress conditions. A line × tester mating design was used to study the lines and testers’ general combining ability (GCA) effects. The specific combining ability (SCA) of the hybrid rice combinations was measured under three different irrigation regimes; 6, 9, and 12 days. The study was carried out at the experimental farm of Sakha Agricultural Research Station, Sakha, Kafr El-Sheikh, Egypt, during the 2018 and 2019 rice growing seasons. Due to the genotypes and their partitions to the parents and the crosses, the mean squares were highly significant for all studied traits under the three irrigation regimes. The additive gene effects play an important role in expressing most of the studied traits. Therefore, the selection procedures based on the accumulation of the additive effect would be successful at improving these traits and the grain yield. The cytoplasmic male sterile (CMS) line G46A (L1) was the best combiner for most yield component traits in the three irrigation regimes. The newly devolved restorer lines T11, T1, T2, T5, T4, and T3, as well as the new hybrids L2 × T10, L2 × T6, L1 × T7, L1 × T5, L1 × T3, L2 × T7, L2 × T9, L2 × T8, L2 × T4, L1 × T4, L2 × T2, L1 × T8, L1 × T9, and L2 × NRL 10, showed good, desirable values of the studied traits such as earliness of flowering, short plant height, number of panicles/plant, panicle length, number of spikelets/panicle, number of filled grains/panicle, panicle weight, 1000-grain weight, hulling percentage, milling percentage, head rice percentage, and grain yield under the irrigation regimes of 6, 9, and 12 days. The hybrids L2 × T10, L2 × T6, L1 × T7, and L1 × T5, showed significant positive SCA effects for grain yield, under all three irrigation regimes.

## 1. Introduction

Amongst cereals, rice (*Oryza sativa* L.) is one of the most important staple crops globally for food security [1,2,3]. Rice is very important in Egypt because of its high economic impact. The rice cultivated area in Egypt is 360.44 thousand hectares. The total rice production in Egypt was evaluated at 3.15 million tons in 2018 [4,5].

Water scarcity is a major global problem. This challenge has a high impact on agriculture. Rice is considered a sensitive crop to water stress, which affects grain yield and quality [6,7]. To overcome the water deficiency problem, there is a demand for developing new hybrids from rice that can tolerate drought conditions with high yield [5]. Serraj (2011) [8] reported that the reduction in irrigated water can be achieved rapidly by developing new varieties that require less irrigation without a decrease in grain yield and quality traits. Subsequently, water productivity can be increased by using promising genotypes that produce high yields in drought-prone areas, certain agronomic practices, or a suitable irrigation regime [9]. High-yield genotypes are essential in different weather conditions, and the cultivation of these genotypes is necessary to keep higher productivity and quality content of rice grain under water stress [10,11,12].

Grain characteristics that meet consumer preferences are part of rice grain quality; the physical properties of rice grains include milling recovery [13,14]. In addition, the physical properties of rice grains differ in different genotypes of rice and affect the quality characteristics of the grain [15].

Several methods and techniques are used to evaluate new rice genotypes regarding their tolerance to water stress conditions at different crop development stages [12,16,17,18]. Field trials have been used as the main criteria for evaluating drought stress scoring methods and selecting stress-tolerant rice cultivars during crop growth stages [19]. Genotypes that produce high yields under water stress conditions have been selected as drought-tolerant genotypes [12]. Several pressure tolerance indices [20,21,22,23,24,25,26,27,28,29,30] have been used for yield production under well water and water pressure conditions. Generally, stress indicators are calculated based on the tolerance to stress or the sensitivity to stress based on a genetic genotype, according to [23,25]. These genotypes can be used as indicators to evaluate water stress effects based on yield losses under stress conditions compared to optimal water conditions [31]. During stress, a decline in yield indicates the susceptibility of the genotype to stress [32]. Fischer and Maurer (1978) [20] suggested that a genotype’s stress sensitivity index (SSI) and tolerance index (TI) explain the difference in productivity between different water stress conditions [21].

Parental lines with good combining abilities are considered the main factor for successful hybrid rice breeding that provides a promising tool for increasing rice production. Therefore, the selection process for developing potential parental lines is always a big challenge for rice breeders. In addition, the combining ability was determined to investigate the ability of a specific parental line to transfer its genetic information to its progeny [33,34,35]. 

Line × tester analysis is an effective biometric tool for measuring the general combining ability (GCA) and specific combining ability (SCA). It can also provide information about the nature of gene actions [36,37].

Therefore, this study’s objective was to evaluate the performance, combining ability, and gene effects for yield-related and grain quality traits among newly developed restorer lines. This goal was achieved through hybridization according to a line × tester mating design under both normal and two water-stress conditions. This could be utilized to develop promising, high-yield hybrid rice varieties.

## 2. Materials and Methods

### 2.1. Plant Materials

The current study was carried out at the experimental farm of Sakha Agricultural Research Station, Sakha, Kafr El-Sheikh, Egypt, (30°57′12″ north latitude, 31°07′19″ east longitude) during the 2018 and 2019 rice growing seasons. The soil’s mechanical properties were clay (64.3%), sand (7.6%), and silt (28.1%), with pH ranging from 7.8 to 8.5, and the weather was hot and humid. The genetic materials consist of 16 rice lines, including two cytoplasmic male sterile (CMS) lines used as female parents, Gang46A (L1) and IR69625A (L2). These two lines were replaced by their maintainer lines during the field evaluation of the parental lines. Eleven new promising rice restorer lines, NRL 2, NRL 9, NRL 10, NRL 11, NRL 12, NRL 29, NRL 42, NRL 43, NRL 44, NRL 47, and NRL 50 (T1, T2, T3, T4, T5, T6, T7, T8, T9, T10, and T11, respectively), in addition to a commercial restorer variety Giza178 (T12), were used as male parents. The control hybrid in this study was the Egyptian Hybrid 1 (L2 × T12), a commercial hybrid in Egypt (Table 1).

### 2.2. Experiments Setup and Crop Management

In the growing season of 2018, hybridization between parents was carried out using a line × tester mating design by hand crossing according to the technique proposed by [38] to produce 24 hybrids (F_1_). In the growing season of 2019, all genotypes, including F_1_ hybrid combinations with their male and female parents, grew in a field trial containing three different experiments. All experiments were designed using a randomized complete block design (RCBD) with three replications. Seedlings at 25 days old were transplanted with one seedling per hill. Spacing of 20 cm between rows and 20 cm between plants was applied. The first experiment was performed under an irrigation regime at 6-day intervals, the second experiment was performed under an irrigation regime at 9-day intervals to model moderate water stress conditions, and the third was performed under an irrigation regime at 12-day intervals to model strong water stress conditions. These conditions were applied using a flash irrigation system. The irrigation system was applied 15 days after transplant and continued until the maturity stage. The cultural practices with the standard recommendation were conducted as recommended by the Rice Research and Training Center (RRTC), Field Crops Research Institute, Agricultural Research Center, Giza, Egypt. Five random plants from the central rows in each replication were selected and evaluated for their yield and component traits.

### 2.3. Studied Traits

Data were recorded on days to 50% heading (day), plant height (cm), panicle length (cm), panicle length (cm), number of panicles per plant, spikelet fertility (%), number of spikelets per panicle, number of filled grains per panicle, panicle weight (g), grain yield per plant (g), 1000-grain weight (g), hulling percentage, head rice recovery percentage, and milling percentage. All the measurement techniques were based on the International Rice Research Institute (IRRI) Standard Evaluation System [19].
(1)$$\mathrm{Yield}\;\mathrm{reduction}\;\mathrm{index}\;\left(\mathrm{YRDI}\right)=\left(1-\left(\frac{\mathrm{Ys}}{\mathrm{Yp}}\right)\right)\times 100$$
where Yp and Ys are grain yields of each genotype under non-stress and stress conditions, respectively [20].

### 2.4. Statistical Analysis

The data were analyzed using the analysis of variances (ANOVA) test for RCBD as suggested by [39] to test the significance of differences among the genotypes. Line × tester (L × T) analysis was performed according to [36]. General combining ability (GCA) and specific combining ability (SCA) effects were estimated according to [40]. The genetic components were estimated based on the expectations of mean squares according to [41]. Principal component analysis (PCA) and two-way hierarchical cluster analysis (HCA) were carried out using JMP Data analysis software, Version 16 (JMP Statistical Discovery LLC, Cary, NC, USA) [42].

## 3. Results

### 3.1. Analysis of Variance

The analysis of variance (ANOVA) for grain yield and their contributing traits as well as some grain quality traits under the irrigation regimes of 6, 9, and 12 days are presented in Appendix A. The results revealed that the mean squares due to genotypes and the partitions to the parents of the crosses were highly significant for all studied traits under the three irrigation regimes. Meanwhile, the mean squares of lines were significantly or highly significantly different under all three irrigation regimes for all studied traits except for the number of panicles per plant under the irrigation regime of 6 and 9 days, panicle length under 12 days, spikelet fertility percentage under 6 and 9 days, and grain yield per plant and hulling percent under all three irrigation regimes. Moreover, the mean squares of testers were significant and highly significant under all three irrigation regimes for all studied traits except for hulling percentage under 9-day irrigation. The mean square of line × tester showed significant and highly significant differences under all three irrigation regimes for all studied traits except panicle length under the irrigation regime of 6 and 9 days (Appendix A). 

The estimation of the ratio between K^2^ GCA (additive gene effects) and K^2^ SCA (non-additive gene effects) for the grain yield and contributing traits as well as grain quality traits under the irrigation regimes of 6, 9, and 12 days are presented in Appendix A. The ratio of K^2^ GCA/K^2^ SCA was more than unity for all studied traits except for days to heading, panicle length, and panicle weight under 12-day irrigation; 1000 grain weight under 6-day irrigation; and hulling percentage under 12-day irrigation (Appendix A).

### 3.2. Effect of Water Stress on Yield Performance and Productivity

The mean performances of the morphological and yield characters of the studied genotypes are presented in Table 2, Table 3 and Table 4 and Appendix A. For days to 50% heading, the data revealed that L1 was the earliest under all three irrigation regimes. The two testers T8 and T9 were the earliest under the irrigation regimes of 6 days, 9 days, and 12 days (Table 2). Among the hybrids, the L1 × T4 hybrid was the earliest under the three irrigation regimes of 6, 9, and 12 days. For plant height, L1 showed the shortest plant height (desirable) and gave the lowest mean value (Table 2). However, the testers T8, T7, and T9, showed the lowest values under the three irrigation regimes (6, 9, and 12 days). While the most desirable mean values for tallness for the restorer lines were found in the restorers T11, T1, and T6, the CMS line L1, and the CMS line L2 (Table 2). The most desirable mean values for shortness were found in the three hybrids; L2 × T9, L2 × T8, and L2 × T4 under the irrigation regimes of 6, 9, and 12 days. Concerning the number of panicles per plant in the three genotypes T10, T4, and T8 gave the highest mean values (Table 2) under the three irrigation regimes of 6, 9, and 12 days. The four hybrids; L2 × T10, L1 × T1, L1 × T11, and L2 × T6, gave the highest mean values under the three irrigation regimes (Table 2). For 1000–grain weight, the highest value was recorded for L1. The four testers, T8, T2, T9, and T7 gave the highest mean values of 1000–grain weight under the three irrigation intervals (Appendix A, Figure 1). The nine hybrids; L1 × T4, L1 × T6, L1 × T7, L1 × T2, L1 × T10, L2 × T7, L1 × T3, L1 × T9, and L2 × T6 gave the highest mean values of the 1000–grain weight under the three irrigation regimes (Appendix A, Figure 1). Regarding the grain yield per plant, the results indicated that the line L2 showed the highest mean value under all irrigation regimes. The four testers T1, T2, T5, and T11 showed the highest mean values under the irrigation regime of 6 and 9 days (Figure 2A). On the contrary, the testers T3, T5, T2, and T4 showed the highest mean values under the irrigation regime of 12 days (Appendix A, Figure 2A).

For the grain quality traits hulling percentage, milling percentage, and head rice percentage, the new hybrids and all newly developed restorer lines (testers) showed the highest mean performance values under all three irrigation regimes. Moreover, the testers T10, T5, and T1, as well as the hybrids L1 × T3 and L1 × T11 showed the highest mean values for grain quality traits under the three irrigation regimes (Appendix A).

With regard to contributing traits and grain yield, T11, T1, T2, T5, T4, T3, L2 × T10, L2 × T6, L1 × T7, L1 × T5, L1 × T3, L2 × T7, L2 × T9, L2 × T8, L2 × T4, L1 × T4, L2 × T2, L1 × T8, L1 × T9, and L2 × T3 recorded good, or desirable values for the traits earliness, shortness, number of panicles/plant, panicle length, number of spikelets per panicle, number of filled grains per panicle, panicle weight, 1000-grain weight, hulling percentage, milling percentage, head rice percentage, and grain yield under the irrigation regimes of 6, 9, and 12 days. 

The female parent showed a yield reduction of 20.7% and 28.67% for L1 and L2, respectively, under the irrigation regime of 9 days, while it showed a yield reduction of 35.4% and 38.3% for L1 and L2, respectively, under the irrigation regime of 12 days (Figure 2B and Appendix A). Concerning the newly developed restorer lines, the yield reduction ranged from 17.7% to 36.66% for T7 and T10, respectively, under the irrigation regime of 9 days, while under the irrigation regime of 12 days it ranged from 31.6% to 57.2% for T3 and T10, respectively (Figure 2B and Appendix A). This result indicates that these lines were subjected to severe water stress. The hybrid check variety (L2 × T12) showed a yield reduction of 36.7% and 51% under the irrigation regimes of 9 and 12 days, respectively. For new hybrids, the yield reduction ranged from 12.6% to 30.4% for hybrid L1 × T9 and hybrid L2 × T10, respectively, under the irrigation regime of 9 days, with a value of 31.1% for Egyptian Hybrid 1 (L2 × T12). On the contrary, the new hybrids showed a yield reduction ranging from 12.6% to 64.4% for hybrid L1 × T9 and hybrid L2 × T10, respectively, under the irrigation regime of 12 days, with a value of 43.6% for Egyptian Hybrid 1 (the check hybrid) as a control (Appendix A and Figure 2B). 

### 3.3. General Combining Ability (GCA) Effects

The general combining ability (GCA) effects for lines and testers under the irrigation regimes of 6, 9, and 12 days are presented in Table 5, Table 6 and Table 7. The data revealed that line L2 gave highly significant negative and desirable values for days to 50% heading and plant height under the three irrigation regimes. Moreover, the same line gave highly significant and significantly positive desirable values of spikelet fertility percentage under all three irrigation regimes. Meanwhile, line L1 showed highly significant positive and desirable values for the number of panicles per plant under the irrigation regimes of 9 and 12 days; panicle length under the irrigation regimes of 6 and 9 days; and number of spikelets per panicle, number of filled grains per panicle, panicle weight, 1000-grain weight, milling percentage, and head rice recovery percentage under all three irrigation regimes (Table 5, Table 6 and Table 7).

For the males, the results showed that only seven, seven, and eight out of 12 testers exhibited significant and highly significant negative combining ability effects for days to 50% heading under irrigation regimes of 6, 9, and 12 days, respectively. The lowest desirable values were recorded by testers T8, T9, T4, and T6 under 6 days, while, under 9 and 12 days the testers T4, T7, T6 and T5, showed the lowest desirable values (Table 5). For plant height, the testers T9, T8, T10, T4, T5, T6, and T3 showed highly significant negative and desirable values of GCA effects under the irrigation regime of 6 days, while the testers T4, T9, T7, T6, T5, T8, and T10 showed highly significant negative and desirable values of GCA effects under the irrigation regime of 9 days (Table 5). Moreover, under an irrigation regime of 12 days, the testers T6, T8, T7, and T9 showed highly significant negative and desirable values of GCA effects. The negative values are indicators of decreased plant height. For the number of panicles per plant, the results showed that the testers T10 and T11 exhibited the highest and highly significant positive values of GCA effects under the three irrigation regimes, respectively. For panicle length, the testers T11 and T12 had highly significant positive desirable values under the irrigation regime of 6 days, while T11 and T10 had highly significant positive desirable values under the irrigation regime of 9 days (Table 5). In addition, six testers had highly significant and significant positive desirable values under the irrigation regime of 12 days, and the highest values were recorded for T11, T10, and T5 (Table 5). 

Regarding the number of spikelets per panicle and the number of filled grains per panicle, the results showed that highly significant positive and desirable values of GCA effects were recorded by tester T7 under the irrigation regime of 6 days and tester T8 under the irrigation regimes of 9 and 12 days (Table 6). For panicle weight, the data indicated that the three testers T7, T9, and T8 showed highly significant positive values of GCA effects under the irrigation regime of 6 days. At the same time, the four testers T7, T8, T9, and T4 had highly significant and significantly positive values of GCA effects under the irrigation regime of 9 days. However, the five testers T8, T3, T9, and T7 gave highly significant positive values of GCA effects under the irrigation regime of 12 days. For spikelet fertility percentage, the results showed that only seven, six, and six out of 12 testers exhibited highly significant positive values of GCA effects under irrigation regimes of 6, 9, and 12 days, respectively (Table 6). T4 recorded the highest value under the irrigation regimes of 6 and 9 days, and T9 did so under the irrigation regime of 12 days. Regarding 1000-grain weight, only three, six, and six out of 12 testers exhibited highly significant positive values of GCA effects under irrigation regimes of 6, 9, and 12 days, respectively. T7 was the best general combiner at the three irrigation intervals, with values of 1.2, 1.6, and 1.5, respectively (Table 7). For grain yield per plant, eight, seven, and four testers showed highly significant and significantly positive values of general combining ability effects under the irrigation regimes of 6, 9, and 12 days, respectively. The testers T7, T2, T4, and T3, showed the highest significant positive values of GCA effects under the irrigation regime of 6 days. The testers T7, T9, T2, and T3 gave the highest significant positive values of GCA effects under the irrigation regime of 9 days. The testers T9, T3, T2, and T4 recorded highly significant positive values of general combining ability effects under the irrigation regimes of 12 days (Table 7).

The results showed that the testers T5, T10, and T11 were the best general combiners under the irrigation regimes of 6, 9, and 12 days, respectively. For milling percentage, the results revealed that the best GCA were recorded for testers T10 for the irrigation regimes of 6 and 9 days and T9 for the irrigation regime of 12 days. For head rice percentage, the results indicated that the five testers T11, T10, T12, T1, and T2 showed highly significant positive values of GCA effects under the irrigation regime of 6 days. At the same time, the four testers T1, T2, T11, and T12 showed highly significant and significantly positive values of GCA effects under the irrigation regime of 9 days. In addition, the four testers T4, T11, T1, and T12 gave highly significant and significant positive values of GCA effects under the irrigation regime of 12 days (Table 7).

### 3.4. Specific Combining Ability (SCA) Effects

Specific combining ability (SCA) effects for the grain yield and contributing traits of the 24 F_1_ hybrids under the irrigation regimes of 6, 9, and 12 days are presented in Table 8, Table 9 and Table 10. The results reveal that only three, four, and seven out of 24 hybrid combinations exhibited highly significant and significant negative desirable values of specific combining ability effects for days to 50% heading under the three irrigation regimes, respectively. The hybrid L1 × T4 was the best combination under the three irrigation regimes (Table 8). Only six out of 24 hybrids had highly significant and significant negative desirable values under the three irrigation regimes for plant height. However, the best combination was shown for hybrids L1 × T6 under the irrigation regime 6 days and L1 × T1 under the irrigation regimes of 9 and 12 days. Concerning the number of panicles per plant, the result revealed that nine, eight, and nine hybrids recorded highly significant and significant positive desirable values under the irrigation regimes of 6, 9, and 12 days, respectively. The hybrids L1 × T1, L2 × T10, L2 × T6, and L1 × T5 showed the highest significant positive values under the irrigation regime of 6 days. The hybrids L1 × T1, L2 × T6, L2 × T10, and L1 × T5 showed the highest significant positive values under the irrigation regime of 9 days. The hybrids L2 × T9, L1 × T1, L1 × T5, and L2 × T10 showed the highest significant positive values under the irrigation regime of 12 days (Table 8). For panicle length, the two hybrids L1 × T1 and L2 × T4 had highly significant positive values under the irrigation regime of 6 days. At the same time, the hybrids L2 × T3, L1 × T7, and L1 × T8 had significant positive values under the irrigation regime of 9 days, while the five hybrids L1 × T7, L1 × T8, L1 × T2, L2 × T4, and L2 × T6 showed significant and highly significant positive values under the irrigation regime of 12 days (Table 8). Concerning the number of spikelets per panicle, nine, nine, and ten hybrids had highly significant (desirable) positive values of SCA effects under the three irrigation regimes, respectively. The hybrids L2 × T6, L1 × T1, and L2 × T6 had the highest values of SCA effects under the irrigation regimes of 6, 9, and 12 days, respectively, (Table 9). Regarding the number of filled grains per panicle, there were 11, 11, and 10 hybrids that exhibited highly significant and significant positive desirable values of SCA effects under the three irrigation regimes, respectively. The highest significant positive values were shown in the hybrids L2 × T6, L2 × T3, and L2 × T4 under the irrigation regimes of 6, 9, and 12 days, respectively (Table 9). For panicle weight, 8, 10, and 7 hybrids had highly significant or significant positive and desirable values of SCA effects under the three irrigation regimes, respectively. The hybrids L1 × T5 and L2 × T11 recorded the highest significant positive values under the irrigation regimes of 6, 9, and 12 days, respectively. Regarding spikelet fertility percentage, the results showed that eight, five, and nine hybrids recorded highly significant and significantly positive (desirable) values of specific combining ability effects under the three irrigation regimes of 6, 9, and 12 days, respectively. The hybrids L2 × T3, L2 × T3, and L2 × T4 had the highest significant positive value of specific combining ability effects under the irrigation regimes of 6, 9, and 12 days, respectively (Table 9).

For 1000-grain weight, the results revealed that only 8, 10, and 7 out of 24 hybrids had highly significant and significantly positive desirable values of SCA effects under the irrigation regimes of 6, 9, and 12 days, respectively. The hybrids L1 × T9 and L2 × T6 were the best combinations under the irrigation regimes of 6, 9, and 12 days (Table 10). For grain yield per plant, 10, 8, and 7 hybrids showed highly significant and significantly positive desirable values of SCA effects under the three irrigation regimes, respectively. The hybrids L1 × T3, L1 × T2, L2 × T6, L2 × T10, and L1 × T5 exhibited the highest significantly positive values under the irrigation regime of 6 days. Additionally, the hybrids L2 × T10, L2 × T6, and L1 × T5 had the highest significantly positive values under the irrigation regime of 9 days, while the hybrids L1 × T7, L2 × T10, L2 × T2, and L1 × T5 exhibited the highest significantly positive values under the irrigation regime of 12 days (Table 10). 

For hulling percentages, 3, 7, and 7 out of 24 hybrids exhibited highly significant and significant positive values of SCA effects under the irrigation regimes of 6, 9, and 12 days, respectively. The best hybrid combinations were L1 × T5, L1 × T1, and L1× T1 under the irrigation regimes of 6, 9, and 12 days, respectively. For milling percentage, the results showed that 7, 8, and 7 hybrids had highly significant and significantly positive values of SCA effects under the three irrigation regimes, respectively. The highest significantly positive values were shown in the hybrids L2 × T12, L2 × T12, L2 × T9 under the irrigation regimes of 6, 9, and 12 days, respectively. Regarding head rice percentage, 10, 8, and 7 hybrids recorded significant positive and desirable values of SCA effects under the three irrigation regimes, respectively. The hybrids L2 × T7, L2 × T7, and L2 × T9 showed the highest significant positive values under the irrigation regimes of 6, 9, and 12 days, respectively (Table 10).

The proportional contributions of lines, testers, and line × tester interactions for the expression of traits is presented in Table 11. It is apparent that the testers were more important under the irrigation regimes of 6, 9, and 12 days for days to 50% heading, plant height (except under the irrigation regime of 9 days), panicle length, number of spikelets per panicle, number of filled grains per panicle, panicle weight, spikelet fertility, 1000-grain weight, grain yield per plant (except under the irrigation regimes of 6and 9 days), hulling percentage (except under the irrigation regime of 9 days), milling percentage (except under the irrigation regimes of 6 and 9 days), and head rice percentage. At the same time, the contribution of lines was more important for plant height (under the irrigation regime of 9 days) and milling percentage (under the irrigation regimes of 6 and 9 days). Additionally, under the irrigation regimes of 6, 9, and 12 days, the contribution of maternal and paternal interaction (line × tester) was more critical for the number of panicles per plant, grain yield per plant (except under the irrigation regime 12 days), and hulling percentage (except under the irrigation regimes of 6 and 12 days), (Table 11).

### 3.5. Interrelationships between Genotypes, Irrigation Regimes, and Traits

Heat-map analysis (two-way hierarchical cluster analysis) data indicate significant differences among genotypes (Figure 3). The data showed a positive correlation in the pairs of traits such as days to 50% heading under the three irrigation regimes of 6, 9, and 12 days, grain yield per plant under 6, 9, and 12 days, and 1000-grain weight under 6, 9, and 12 days (Figure 3). The heat-map analysis showed that every trait under different irrigation regimes was in a sub-group. At the same time, the analysis showed that with the presence of close traits within one group with the presence of four main groups, the traits within each group are related to each other and affect each other such as grain yield per plant with 1000-grain weight and spikelet fertility percentage. Heat-map analysis located the genotypes under study in different groups based on the genetic background of the genotypes (Figure 3). According to the heat-map analysis, the genotypes under study have been divided into five clusters. The first cluster has two groups. Group A includes L1, L2, L2 × T1, and L1 × T6, while group B includes T7, T8, and T9. The second cluster also includes two groups. The first group consists of T2, T4, T3, T5, and T10, while the second group includes T6, L2 × T12 (check variety), L2 × T5, L2 × T10, and T12. The third cluster includes the eight genotypes T1, L1 × T1, T11, L1 × T12, L1 × T11, L2 × T12, L1 × T3, and L1 × T10. The fourth cluster consisted of the seven genotypes L1 × T2, L1 × T5, L2 × T4, L1 × T4, L1 × T9, L1 × T7, and L1 × T8, while the fifth and last cluster included the six hybrids L2 × T2, L2 T3, L2 × T8, L2 × T9, L2 × T6, and L2 × T7.

The principal component analysis (PCA) and two-way hierarchical cluster analysis (HCA) of individual response variables in rice hybrids cultivated under three irrigation regimes (of 6, 9, and 12 days) indicated that the traits studied and the percent contribution of different components (lines, testers, and lines × testers) were clustered separately into three distinct clusters (Figure 4A,B). The PCA analysis showed a very good separation of the lines, testers, and hybrids under the three irrigation regimes, while the testers were more correlated with the measured parameters than the lines and the newly developed hybrids (Figure 4A). This was very clear in the HCA analysis since the contributions of the lines were clustered together regardless of the irrigation regime. The contributions of the line × tester hybrids were clustered together as an intermediate group regardless of the irrigation regime. Finally, the contributions of the testers were clustered as one group regardless of the irrigation regime as well.

## 4. Discussion

The parental lines and their hybrid combinations recorded better mean performances in all studied traits. However, the genotypes studied were affected by the irrigation regime in all studied traits (Table 2, Table 3, Table 4 and Appendix A; Figure 1 and Figure 2A,B). Thus, we can use some of these genotypes as new restorer lines, new varieties, new hybrids, and a source for developing new restorer lines and new hybrids for water deficiency tolerance in rice breeding programs under Egyptian conditions.

The main objective of plant breeders when developing new lines or hybrids is to obtain high-yield genotypes that can also provide stable performance against the limitations of growing crops [43]. Therefore, genotypes tolerant to water stress with high yield ability under normal irrigated and water shortage conditions are highly valued. These genotypes (lines or hybrids) are spread in different areas to save water. Various studies have been carried out to overcome water stress in rice by breeding tolerant varieties using traditional breeding varieties [44]. 

Based on the results of previous studies, Awad-Allah (in 2006 and 2011) [45,46] found that the parental lines T12, BG 33-5, and BG 34-8 have the *Rf1* allele where the M2, a dominant marker linked to the *Rf1* gene, on chromosome 1. Additionally, these lines were identified as tolerant to water shortages (drought). Moreover, [45,46] revealed a band detected by RM 171 marker in studied genotypes, which suggests that these lines may have the Rf4 allele that is known to be linked with the RM 171 marker on chromosome 10 in WA CMS lines.

Based on the findings of molecular and field analyses by Awad-Allah in 2011 [46], the promising hybrids L2 × T12, L1 × BG 33-5, L1 × BG 34-8, and L2 × BG 34-8 were grow to produce F_2_ and the selection involved F_2_ to F_7_. The promising lines were selected and grown along with the parental lines and evaluated for phenotypic performance and yield ability. Therefore, the newly developed restorer lines contain restorer genes from the parents.

In this study, new lines have been developed by crossing two cytoplasmic male sterile lines with three genotypes. Subsequent selections of the lines under normal conditions with some water shortage over generations were conducted since the well-watered condition is the primary target in rice breeding and, as in most cases, high-yield lines can still give high to moderate yields under water shortage conditions [43,47,48,49].

It can be concluded that water stress delays the flowering time for most of the parents (the newly developed lines) and some hybrids. This has previously been observed in [50,51]. Some studies have indicated that delayed flowering under water stress is a good predictor of plant response and may explain plants’ adaptability to water deficiency [52]. Under different irrigation regimes, the differences in heading dates for the same genotype can be attributed to the extended vegetative stage due to water stress. Lafitte et al. (2004) [53] reported that water deficit results in delayed heading. This is mainly due to a reduction in plant dry matter production and slowed elongation of the panicle and supporting tissues and therefore delayed panicle excretion. At the same time, water stress causes earlier flowering in female parents and controls. In the current study, water-shortage stress negatively affects the performance of the newly developed restorer lines and the hybrids resulting from it. A negative effect has previously been reported by [44,54], in which water shortage caused a severe reduction in plant height, spikelet fertility, grain yield, and its components. Water stress pushes plants to perform more respiration and reduce photosynthesis, leading to less biomass accumulation and less grain yield [55]. Plant height was decreased with increased intervals up to 12 days. The reduction in plant height could be attributed to a decrease in cell size that causes a decrease in cell swelling, which decreases shoot development. Ahmed et al. (2017) [56] reported that water stress strongly influences plant growth and decreases plant height. Water stress reduces the cell size and division, affecting plant height under water stress. However, the reduction in the number of panicles per plant could be attributed to a lack of ability of tiller nodes to produce more tillers under water stress. A similar trend was found in [7,57,58].

Water shortage stress caused a reduction in the number of panicles per plant, panicle length, number of spikelets per panicle, spikelet fertility percentage, number of filled grains per panicle, panicle weight, 1000-grain weight, grain yield per plant, hulling percentage, milling percentage, and head rice recovery percentage. These findings are in harmony with those obtained by [59], who reported that drought stress caused many constructional and functional disruptions in floral organs, leading to malfunctions in fertilization or premature abortion of the seed, a decreased grain filling period, early senescence, photosynthesis reduction, and enhanced soluble sugar remobilization from grains to other vegetative parts, which were observed when water stress happened at the reproductive stage. Sugar or carbohydrate remobilization depends on source activity and sink strength, which varies by genotype. This may be because the growth characteristics under the conditions of shorter irrigation regimes of 6 and 9 days are better (increased dry matter, increased chlorophyll content, and increased plant height) and are associated with a higher ability to absorb mineral nutrients from the soil, which leads to increased absorption of nutrients and contributes to increased growth, thus resulting in a higher yield.

The female parent showed yield reductions of 20.7% and 28.67% for L1 and L2, respectively, under the irrigation regime of 9 days and yield reductions of 35.4% and 38.3% for L1 and L2, respectively, under the irrigation regime of 12 days. Several previous studies have reported more than 50% yield reductions, confirming successful water stress screening [60]. A reduction in panicle weight, panicle length, number of panicles per plant, number of spikelets per panicle, number of filled grains per panicle, spikelet fertility, and 1000-grain weight are some of the causes of grain yield reduction under water stress [35,61,62,63].

These results agree with those of [58,64], who stated that such increments in yield attributes under non-stress conditions could be due to the fact that available water enhances the biological and physiological processes, which increases the production and translocation of the dry matter content from source to sink, resulting in more panicles, greater grain filling, and greater grain weight. 

The current research results confirm the possibility of using grain yield under water deficiency as an effective direct selection criterion for enhancing rice tolerance to water shortage stress. However, this study includes newly developed restorer lines (inbred lines) using a common parent. The new hybrids resulting from these lines showed highly significant values for the studied traits under the three irrigation regimes. This indicates that the presence of significant variations between the different genotypes.

The analysis of variance for various genotypes (restorer lines and hybrids) at three irrigation regimes showed significant differences for all studied traits; this indicates the presence of large variations between genotypes. Thus, the genotypes evaluated can be selected for genetic improvement for grain yield and other agronomic traits under water shortage conditions. Previous researchers have emphasized the importance of genetic variation in breeding new improved varieties [65,66]. The analyses of variance of line × tester showed significant and highly significant differences under the three irrigation regimes for all studied traits except panicle length under 6 days and 9 days. Similar results were obtained in [54,67,68,69,70,71].

Genetically, the GCA indicates additive and additive × additive gene action; on the other hand, the SCA detect the non-additive gene action. The results indicate that the preponderance of additive genes affect the expression of all studied traits (except days to heading, panicle length, panicle weight under 12 days irrigation regime, 1000 grain weight under 6 days irrigation regime, and hulling percentage). Therefore, selection procedures based on the accumulation of additive effects would successfully improve grain yield and its contributing traits. These results agree with the results obtained in [35,68,69,71].

In the current study, the two CMS lines performed the best for some yield–component traits under the irrigation intervals 6 days, 9 days, and 12 days. The CMS line L1 (G46A) was the best combiner for most of the yield component traits under the irrigation regimes of 6, 9, and 12 days. This finding was promising compared to those of previous studies. Past studies have indicated that no particular parent possessed the best GCA effects [71,72]. In contrast, most restorer lines recorded desirable values of the GCA effect for the studied traits. Among these, the newly devolved restorer lines T11, T1, T2, T5, T4, and T3 showed good, desirable values of the studied traits such as earliness, shortness, number of panicles per plant, panicle length, number of spikelets per panicle, number of filled grains/panicle, panicle weight, 1000-grain weight, hulling percentage, milling percentage, head rice percentage, and grain yield under the irrigation regimes of 6, 9, and 12 days. Additionally, a higher GCA being found in the restorer lines has been reported in former studies [35,67,70,73]. This means that these genotypes could be utilized as parents for breeding genotypes with more fertile grains per panicle for developing maintainers, restorer lines, and hybrids; these findings agreed with those of [35,46].

The values of SCA effects of the hybrid combinations indicated that all the hybrids present significant positive values of SCA effects for at least one yield component. Overall, the results suggested that no specific combination had positive SCA values for all traits in this study. It can also be concluded that the best hybrid combinations with a highly significant SCA for various traits had at least one parent with a good GCA. For days to 50% heading, the hybrid L1 × T4 was the best combination under the three irrigation regimes and exhibited a highly significant negative value. This hybrid appears to be a good specific combiner to develop early restorer lines and hybrids. However, the best combination for plant height was shown for hybrid L1 × T6 under the irrigation regime of 6 days and L1 × T1 under the irrigation regimes of 9 and 12 days. Negative values of SCA effects mean a decreased plant height and this could be helpful for breeding short-stature rice hybrids and rice cultivars, while positive values of SCA effects mean increased plant height and this could be helpful for breeding restorer lines. These results agree with the results obtained in [35,46]. In addition, the hybrids L2 × T10, L2 × T6, L1 × T7, and L1 × T5 showed significant positive SCA effects for grain yield under the irrigation intervals of 6 days, 9 days, and 12 days. These genotypes appeared to be good combiners to improve rice cultivars and hybrids for grain yield/plant under water shortage. These findings agree with the results obtained in [35,46,67,72,74]. The hybrid L1 × T5 had the highest SCA for the grain yield among hybrids whose parents had a negative GCA for that trait under the irrigation regimes of 6, 9, and 12 days (Table 10). At the same time, the hybrid IR69625 × T6 showed a desirable value of SCA for grain yield, while one of the parents showed negative GCA for that trait under the irrigation regimes of 6, 9, and 12 days. In contrast, the hybrid L1 × T7 showed negative SCA effects despite its parents having good GCAs for the 1000-grain weight under the irrigation regimes of 6, 9, and 12 days. These findings agree with the results obtained in [35,75]. These results probably occurred due to interactions between the positive and negative alleles of the parents’ genes and their complex combinations. This may be due to a poor combiner parent producing epistatic effects and a good combiner parent displaying suitable additive effects [76,77]. These findings were similar to the findings of previous studies [71].

The current study revealed that the testers were more important under all irrigation intervals for days to 50% heading, plant height (except under the irrigation regime of 9 days), panicle length, number of spikelets per panicle, number of filled grains per panicle, panicle weight, spikelet fertility, 1000-grain weight, grain yield per plant (except under the irrigation regimes of 6 and 9 days), hulling percentage (except under the irrigation regime of 9 days), milling percentage (except under the irrigation regimes of 6 and 9 days), and head rice percentage. These results are similar to those obtained by [77] for days to 50% flowering and plant height and [75] for number of spikelets per panicle. Bare preponderance testers influence these traits. At the same time, the contribution of lines was more important for other traits. These results agree with the results obtained in [78] for 1000-grain weight. In [77,78,79,80,81], they found that the contribution of maternal and paternal interaction (line × tester) was more important for grain yield per plant.

## 5. Conclusions

The newly developed restorer lines; T11, T1, T2, T5, T4, T3, as well as their new hybrids; L2 × T10, L2 × T6, L1 × T7, L1 × T5, L1 × T3, L2 × T7, L2 × T9, L2 × T8, L2 × T4, L1 × T4, L2 × T2, L1 × T8, L1 × T9, and L2 × T3 gave good, desirable performance values for the traits earliness, shortness, number of panicles per plant, panicle length, number of spikelets per panicle, number of filled grains per panicle, panicle weight, 1000-grain weight, hulling percentage, milling percentage, head rice percentage, and grain yield under normal irrigation regimes of 9 days and 12 days. The hybrids; L2 × T10, L2 × T6, L1 × T7, and L1 × T5 showed significant positive SCA effects for the grain yield under the three irrigation regimes of 6, 9, and 12 days. The newly developed restorer lines; T7, T8, T1, and T9 showed the lowest reduction in grain yield, while the check variety (control) showed a yield reduction of 38.2% under the irrigation regime of 9 days. On the contrary, the newly developed restorer lines; T3, T6, T9, and T5 showed the lowest reduction in grain yield compared to the control variety, which showed a yield reduction of 51% under the irrigation regime of 12 days. These genotypes appear to be good combiners to improve and release rice cultivars and hybrids to increase grain yield per plant under water-deficient conditions.

## Figures and Tables

**Figure 1 genes-13-00906-f001:**
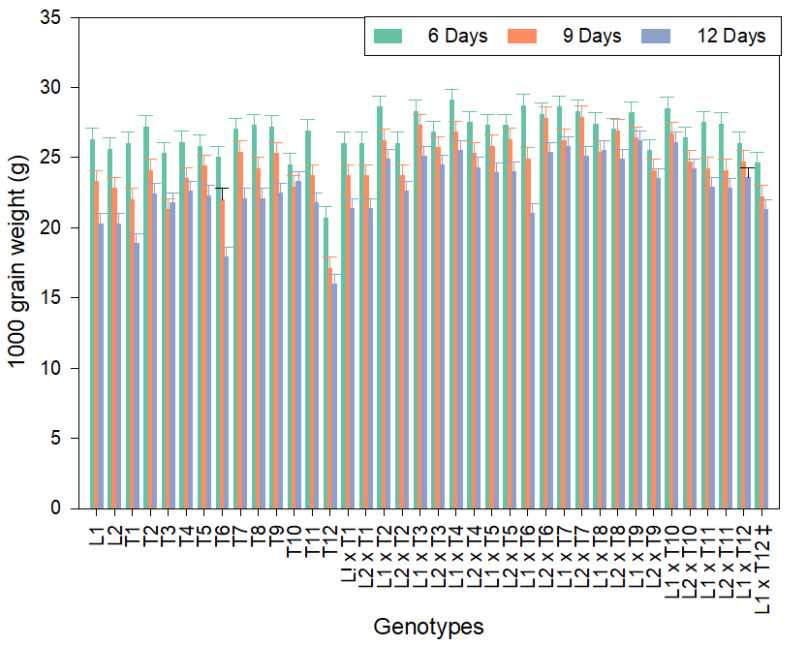
The performance of 1000-grain weight for the newly developed hybrids and their parental rice lines under the three irrigation regimes (6, 9, and 12 days). ^‡^: control hybrid. Bar represents the LSD at 5%; 0.8, 0.8, and 0.7 for the 6-, 9-, and 12-day irrigation regimes, respectively.

**Figure 2 genes-13-00906-f002:**
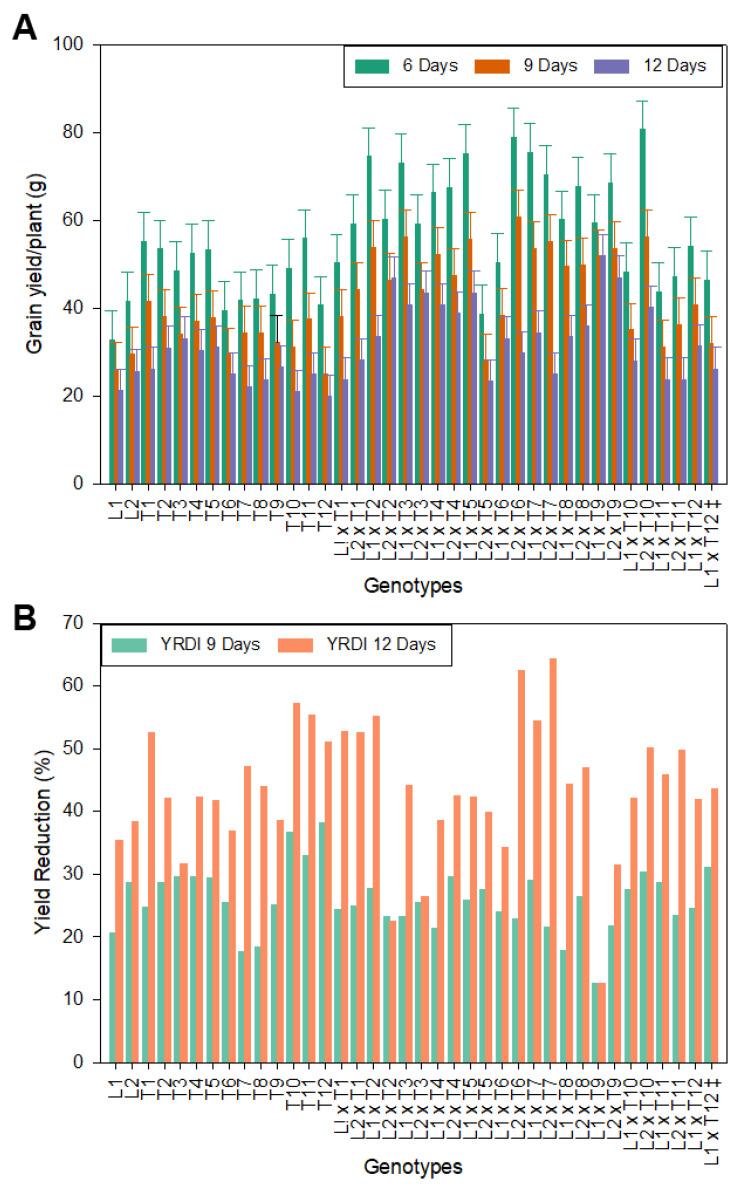
The performance of grain yield per plant (**A**) and yield reduction (**B**) for the newly developed hybrids and their parental lines of rice under three irrigation regimes; 6, 9, and 12 days. YRDI: yield reduction index. ^‡^: control hybrid. Bar represents the LSD at 5%; 6.6, 6.1, and 4.9 for the 6-, 9-, and 12-day irrigation regimes, respectively.

**Figure 3 genes-13-00906-f003:**
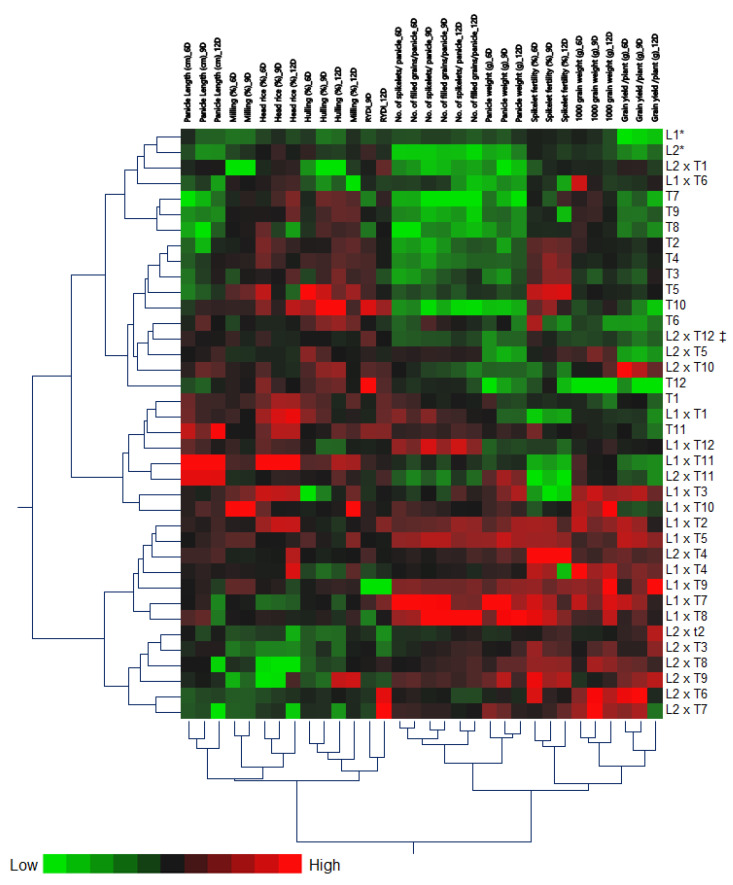
Two-way hierarchical cluster analysis (HCA) of individual response variables assessed in rice hybrids cultivated under three different irrigation regimes of 6, 9, and 12 days. Variations in the dependent variables among the studied rice hybrids are visualized as a heat map. Dependent variables are presented in rows, while different treatments are presented in the column. The map presents high numerical values in red and low numerical values in green (the scale is at the bottom left corner of the heat map). *: Line cultivar and ^‡^: the check verity.

**Figure 4 genes-13-00906-f004:**
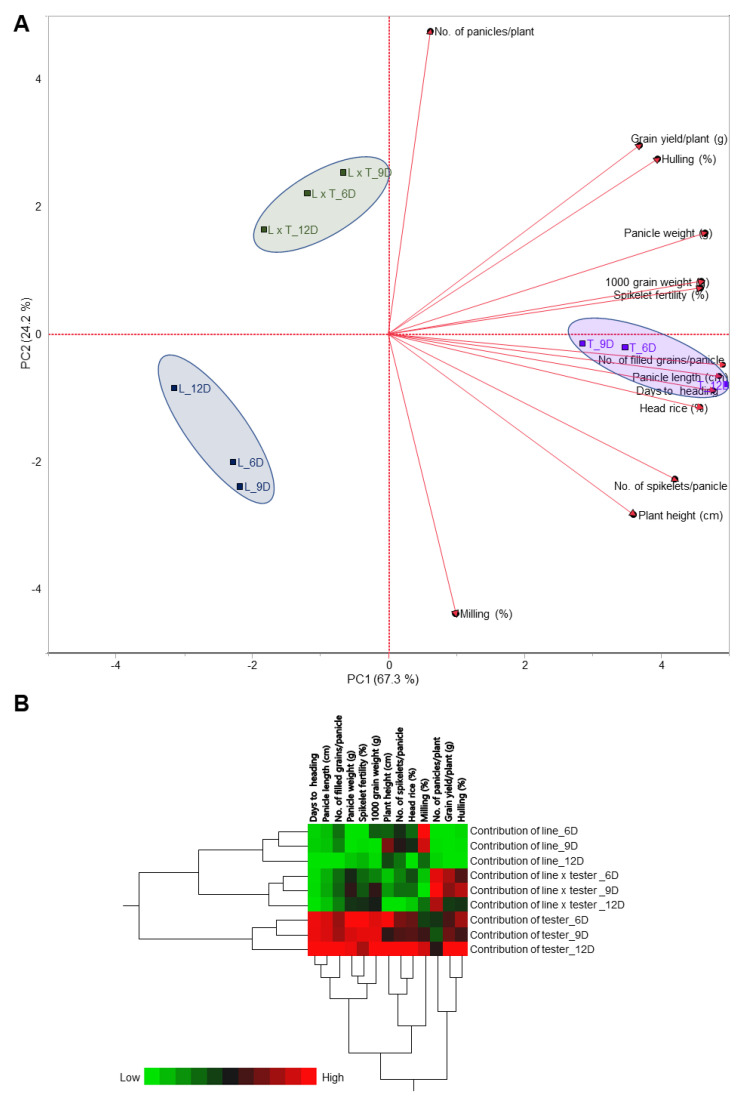
Principal component analysis (PCA) and two-way hierarchical cluster analysis (HCA) of individual response variables in the tested rice hybrids cultivated under three irrigation regimes (6, 9, and 12 days). (**A**) PCA-associated scatters and PCA-associated loading plots. (**B**) Two-way HCA. Variations in the dependent variables between studied treatments are presented as a heat map. Dependent variables are presented in rows, while different treatments are presented in columns. The map presents high numerical values in red and low numerical values in green (the scale is at the bottom left corner of the heat map).

**Table 1 genes-13-00906-t001:** Hybrid names, codes, and parentage of the studied genotypes.

Female Parents	Code	Male Parents (Parentage)	Code	Hybrids
Gang46A (G46A) Parentage (Erjiu’ai 7/V41B//Zhenshan 97/Ya’aizao)	L1	NRL 2 (IR69A/Giza178)	T1	L1 × T1
		NRL 9 (IR69A/Giza178)	T2	L1 × T2
		NRL 10 (IR69A/Giza178)	T3	L1 × T3
		NRL 11 (IR69A/Giza178)	T4	L1 × T4
		NRL 12 (IR69A/Giza178)	T5	L1 × T5
		NRL 29 (G46A/BG33-5)	T6	L1 × T6
		NRL 42 (G46A/BG33-5)	T7	L1 × T7
		NRL 43 (G46A/BG33-5)	T8	L1 × T8
		NRL 44 (G46A/BG33-5)	T9	L1 × T9
		NRL 47 (IR69A/ BG34-8)	T10	L1 × T10
		NRL 50 (G46A/BG34-8)	T11	L1 × T11
		T12 (Giza175/Milyang 49)	T12	L1 × T12
IR69625A (IR69A)	L2	NRL 2 (IR69A/Giza178)	T1	L2 × T1
		NRL 9 (IR69A/Giza178)	T2	L2 × T2
		NRL 10 (IR69A/Giza178)	T3	L2 × T3
		NRL 11 (IR69A/Giza178)	T4	L2 × T4
		NRL 12 (IR69A/Giza178)	T5	L2 × T5
		NRL 29 (G46A/BG33-5)	T6	L2 × T6
		NRL 42 (G46A/BG33-5)	T7	L2 × T7
		NRL 43 (G46A/BG33-5)	T8	L2 × T8
		NRL 44 (G46A/BG33-5)	T9	L2 × T9
		NRL 47 (IR69A/ BG34-8)	T10	L2 × T10
		NRL 50 (G46A/BG34-8)	T11	L2 × T11
		Giza178 (Giza175/Milyang 49)	T12	L2 × T12 *

* Control hybrid.

**Table 2 genes-13-00906-t002:** Mean performance for days to heading, plant height, and the number of panicles/plant of the studied genotypes.

Traits Genotypes	Days to 50% Heading			Plant Height (cm)			No. of Panicles/Plant		
	6 D	9 D	12 D	6 D	9 D	12 D	6 D	9 D	12 D
L1 *	80.2	79.0	77.9	88.7	81.0	73.4	17.6	16.4	14.7
L2 *	100.1	98.7	98.0	104.3	87.7	72.4	19.2	17.6	15.2
T1	103.0	104.3	105.8	119.6	97.2	85.8	15.2	16.3	16.2
T2	101.0	102.7	104.4	106.4	93.0	81.3	20.0	17.3	18.5
T3	102.0	103.1	104.7	110.0	92.8	88.2	18.2	17.7	16.8
T4	97.0	98.8	100.0	110.9	94.2	88.0	21.2	19.6	18.8
T5	99.4	101.5	103.4	97.3	85.7	79.8	19.3	19.3	14.5
T6	90.0	91.1	92.1	115.9	102.8	97.5	17.5	16.5	15.3
T7	90.6	91.6	92.3	92.3	79.8	74.7	19.7	18.7	18.3
T8	87.0	87.8	88.8	90.7	77.7	75.2	20.2	21.3	15.8
T9	87.0	87.8	88.8	92.8	78.2	74.2	17.0	19.1	15.3
T10	101.0	102.3	103.8	103.9	81.3	69.0	21.3	18.0	21.3
T11	105.0	106.0	107.2	138.9	91.3	101.5	19.7	18.3	16.7
T12	101.6	99.8	97.9	104.2	84.8	80.5	18.5	16.4	14.6
L1 × T1	103.0	101.9	100.4	112.2	101.9	98.2	28.5	26.4	21.8
L2 × T1	101.8	100.7	99.5	108.2	102.5	98.9	14.7	13.9	12.3
L1 × T2	102.2	100.7	99.5	105.8	100.6	97.0	19.9	18.8	17.7
L2 × T2	101.4	100.3	99.2	91.8	87.3	83.8	20.3	19.0	17.7
L1 × T3	101.1	100.0	98.6	103.9	99.2	94.0	21.7	21.0	19.7
L2 × T3	100.4	98.7	97.5	89.7	85.0	79.2	23.0	22.0	18.9
L1 × T4	98.4	97.4	96.0	99.8	95.5	91.8	22.3	21.2	18.7
L2 × T4	99.8	98.5	97.2	86.4	80.0	79.6	22.1	20.9	19.5
L1 × T5	100.6	99.4	98.1	101.2	96.3	92.4	23.7	22.6	21.0
L2 × T5	99.8	98.1	96.1	88.5	83.6	79.5	16.0	14.0	11.5
L1 × T6	100.2	99.0	97.2	92.4	89.3	77.5	17.8	16.2	20.3
L2 × T6	99.2	97.9	96.6	98.6	89.7	66.3	24.6	23.7	13.5
L1 × T7	101.0	99.1	97.1	106.5	96.5	83.3	22.8	21.3	12.1
L2 × T7	100.0	98.1	96.1	92.2	81.7	74.4	22.5	21.0	12.3
L1 × T8	99.1	100.4	101.8	98.0	98.2	84.7	19.7	18.5	16.0
L2 × T8	97.6	99.0	101.1	83.8	83.7	70.7	21.4	20.5	18.3
L1 × T9	99.4	100.7	102.6	95.8	97.0	85.0	18.4	16.5	11.8
L2 × T9	98.0	99.6	101.7	78.8	78.7	77.5	21.7	20.9	22.2
L1 × T10	102.1	100.3	98.3	92.0	92.5	86.1	20.6	19.2	17.6
L2 × T10	101.4	99.6	97.9	93.0	89.4	84.6	32.7	26.7	23.7
L1 × T11	106.4	107.9	109.2	120.2	108.1	103.4	28.3	26.1	22.1
L2 × T11	104.6	105.2	106.2	105.4	94.7	91.6	22.9	21.6	18.9
L1 × T12	103.4	101.2	99.9	124.8	98.0	91.0	24.5	23.3	21.4
L2 × T12 ^‡^	102.2	99.6	98.2	108.4	90.6	84.6	20.8	19.2	17.1
L.S.D. 5%	0.72	0.4	0.4	4.41	4.93	3.93	2.0	2.3	1.9
L.S.D. 1%	0.96	0.6	0.5	5.86	6.54	5.21	2.7	3.0	2.6

L.S.D.: least significant difference. ^‡^: control hybrid. L1 * and L2 *: data recorded on the maintainer line; 6 D: 6 days, 9 D: 9 days, 12 D: 12 days.

**Table 3 genes-13-00906-t003:** Mean performance of grain yield and its contributing traits for the studied genotypes.

Traits Genotypes	Panicle Length (cm)			No. of Spikelets/Panicle			No. of Filled Grains/Panicle		
	6 D	9 D	12 D	6 D	9 D	12 D	6 D	9 D	12 D
L1 *	23.5	20.3	18.3	174.2	156.1	120.7	162.0	140.3	103.0
L2 *	22.6	19.5	17.6	132.5	118.8	91.7	123.1	105.7	77.2
T1	26.6	23.1	21.1	238.5	170.8	140.3	220.9	151.2	120.5
T2	22.1	18.9	19.5	143.8	122.4	114.3	137.5	115.5	105.1
T3	21.8	21.0	20.4	143.6	139.8	124.7	136.7	133.1	116.6
T4	22.3	21.1	20.3	151.6	127.5	109.6	144.5	121.6	101.2
T5	21.6	20.4	19.2	170.5	147.0	118.3	166.1	142.9	114.6
T6	23.6	23.8	20.0	165.8	202.3	146.4	161.1	171.4	117.5
T7	20.0	18.9	18.1	141.3	113.4	83.8	131.8	97.1	75.4
T8	20.9	18.3	18.4	126.2	135.1	95.9	116.6	120.7	82.2
T9	21.3	19.8	17.6	153.0	122.7	100.5	141.1	110.1	78.7
T10	23.0	22.9	20.8	150.0	114.8	86.2	143.4	109.7	76.5
T11	28.4	24.3	24.7	222.3	194.9	169.1	213.5	174.5	147.2
T12	22.6	20.3	19.7	192.6	165.5	135.1	174.1	151.3	106.0
L1 × T1	26.7	23.1	20.8	273.6	223.8	161.8	233.6	183.1	128.4
L2 × T1	23.5	22.2	20.2	153.5	120.0	100.2	137.9	106.8	86.4
L1 × T2	25.0	22.4	21.3	248.8	211.4	197.2	241.2	202.6	179.0
L2 × T2	24.2	20.8	19.8	187.7	159.5	148.8	176.4	148.0	134.3
L1 × T3	24.7	21.7	21.1	194.1	173.4	173.7	169.4	136.1	135.1
L2 × T3	23.4	22.4	20.4	186.9	182.0	162.3	180.9	170.7	147.3
L1 × T4	24.5	22.7	20.3	204.7	172.0	150.2	199.1	165.0	117.5
L2 × T4	25.4	23.1	21.6	211.2	177.6	154.7	207.8	174.2	151.9
L1 × T5	24.1	22.8	21.4	278.7	239.8	192.8	270.3	229.7	180.8
L2 × T5	24.1	22.4	21.3	219.9	188.8	151.5	204.4	173.4	136.9
L1 × T6	22.3	21.1	17.3	180.9	161.8	102.4	168.0	142.4	81.1
L2 × T6	22.3	20.8	18.9	209.8	169.0	122.0	205.5	155.6	107.0
L1 × T7	24.1	22.8	17.7	325.8	261.8	187.2	318.3	251.3	168.9
L2 × T7	22.4	20.7	16.4	232.5	186.6	137.6	218.9	172.6	126.6
L1 × T8	25.9	23.9	17.9	282.3	262.6	228.4	276.7	254.1	213.0
L2 × T8	24.1	22.0	16.6	200.6	187.4	163.1	194.0	179.0	153.3
L1 × T9	24.2	22.6	19.0	270.6	217.0	183.4	261.7	204.5	171.2
L2 × T9	23.5	21.9	19.4	246.0	197.0	160.8	241.4	189.0	153.4
L1 × T10	24.7	23.6	21.2	250.9	191.8	143.9	226.2	171.9	124.6
L2 × T10	24.8	24.0	21.8	203.3	155.5	116.7	180.0	137.0	94.4
L1 × T11	29.5	26.4	24.8	190.6	167.1	145.0	164.8	137.0	114.0
L2 × T11	29.0	26.0	24.3	168.7	148.1	128.6	142.9	119.0	99.1
L1 × T12	26.4	23.3	20.3	280.7	250.5	214.5	256.3	219.3	175.7
L2 × T12 ^‡^	25.1	22.2	20.2	167.4	149.8	128.3	158.5	134.5	107.8
L.S.D. 5%	1.4	1.5	1.1	19.5	14.8	9.2	19.5	12.5	8.7
L.S.D. 1%	1.8	2.0	1.5	25.9	19.6	12.2	25.9	16.5	11.5

L.S.D.: least significant difference. ^‡^: control hybrid. L1 * and L2 *: data recorded on the maintainer line; 6 D: 6 days, 9 D: 9 days, 12 D: 12 days.

**Table 4 genes-13-00906-t004:** Mean performances of grain yield and contributing traits for the studied genotypes.

Traits Genotypes	Panicle Weight (g)			Spikelet Fertility (%)		
	6 D	9 D	12 D	6 D	9 D	12 D
L1 *	4.6	3.5	2.9	93.1	89.9	85.3
L2 *	3.6	2.8	2.3	93.1	89.1	84.2
T1	6.6	3.7	2.7	92.6	88.8	86.0
T2	4.4	2.9	2.6	95.6	94.4	91.9
T3	4.0	3.0	2.7	95.2	95.2	93.5
T4	4.3	3.4	2.6	95.3	95.4	92.4
T5	4.9	3.6	2.8	97.4	97.2	96.8
T6	4.6	4.2	2.3	97.1	84.7	80.2
T7	4.1	2.8	1.7	93.3	85.6	90.0
T8	3.7	3.3	2.2	92.5	89.2	85.7
T9	4.2	3.4	2.2	92.3	89.7	78.4
T10	3.4	2.7	2.0	95.7	95.6	88.9
T11	5.7	4.8	3.6	96.1	89.7	87.0
T12	3.1	2.9	2.3	90.4	91.4	78.5
L1 × T1	5.5	3.4	2.5	85.4	81.8	79.3
L2 × T1	3.9	2.5	1.9	89.9	89.3	86.4
L1 × T2	7.0	5.5	5.0	97.0	95.9	90.8
L2 × T2	4.9	3.9	3.4	94.0	92.8	90.3
L1 × T3	6.4	5.6	5.1	87.3	78.6	77.8
L2 × T3	5.6	4.3	3.9	96.8	93.8	90.8
L1 × T4	6.1	4.7	3.4	97.3	95.9	78.4
L2 × T4	6.4	5.3	4.0	98.4	98.2	98.2
L1 × T5	8.2	6.3	5.0	96.9	95.8	93.8
L2 × T5	3.8	2.8	2.1	92.9	91.8	90.3
L1 × T6	3.5	3.1	2.1	92.9	88.2	79.3
L2 × T6	5.1	4.5	3.0	97.8	92.2	87.7
L1 × T7	9.8	6.8	5.0	97.7	96.0	90.3
L2 × T7	7.9	5.4	3.6	94.2	92.6	92.0
L1 × T8	7.6	6.7	5.8	98.0	96.8	93.3
L2 × T8	6.1	5.3	4.3	96.7	95.5	93.8
L1 × T9	7.0	5.7	4.8	96.7	94.3	93.5
L2 × T9	7.4	6.2	3.9	98.1	95.9	95.4
L1 × T10	6.4	5.1	3.7	90.1	89.7	86.7
L2 × T10	4.3	3.4	2.5	88.5	88.1	80.9
L1 × T11	4.5	3.9	3.0	86.5	82.1	78.7
L2 × T11	7.1	6.0	4.6	84.7	80.4	77.0
L1 × T12	4.3	3.7	2.7	91.3	87.6	81.9
L2 × T12 ^‡^	3.8	3.4	2.5	94.7	89.8	84.0
L.S.D. 5%	0.9	0.8	2.4	3.0	4.1	3.4
L.S.D. 1%	1.3	1.0	3.2	4.0	5.4	4.5

L.S.D.: least significant difference. ^‡^: control hybrid. L1 * and L2 *: data recorded on the maintainer line; 6 D: 6 days, 9 D: 9 days, 12 D: 12 days.

**Table 5 genes-13-00906-t005:** General combining ability effects of the parental lines for days to 50% heading, plant height, no. of panicles/plant and panicle length traits.

Traits Parents	Days to 50% Heading			Plant Height (cm)			No. of Panicles/Plant			Panicle Length (cm)		
	6 D	9 D	12 D	6 D	9 D	12 D	6 D	9 D	12 D	6 D	9 D	12 D
L1	0.45 **	0.52 **	0.49 **	5.33 **	5.26 **	4.74 **	0.24	0.33 *	0.60 **	0.43 **	0.32 **	0.09
L2	−0.45 **	−0.52 **	−0.49 **	−5.33 **	−5.26 **	−4.74 **	−0.24	−0.33 *	−0.60 **	−0.43 **	−0.32 **	−0.09
L.S.D. 5%	0.09	0.05	0.04	0.52	0.58	0.46	0.24	0.27	0.23	0.16	0.17	0.13
L.S.D. 1%	0.12	0.07	0.06	0.74	0.82	0.66	0.34	0.38	0.32	0.23	0.25	0.19
T1	1.4 **	1.2 **	0.5 **	11.2 **	9.7 **	12.9 **	−0.5	−0.5	−0.7 *	0.4	−0.1	0.3
T2	0.8 **	0.4 **	−0.1	−0.3	1.5 *	4.8 **	−2.0 **	−1.7 **	−0.1	−0.2	−1.1 **	0.4 *
T3	−0.2 *	−0.8 **	−1.4 **	−2.3 **	−0.4	1.0	0.2	0.9 *	1.5 **	−0.7 **	−0.7 **	0.6 **
T4	−1.9 **	−2.2 **	−2.8 **	−5.9 **	−4.7 **	0.1	0.1	0.4	1.4 **	0.2	0.2	0.8 **
T5	−0.8 **	−1.4 **	−2.3 **	−4.2 **	−2.5 **	0.3	−2.3 **	−2.3 **	−1.5 **	−0.7 **	−0.1	1.2 **
T6	−1.2 **	−1.7 **	−2.5 **	−3.6 **	−3.0 **	−13.7 **	−0.9 **	−0.7 *	−0.8 **	−2.4 **	−1.8 **	−2.0 **
T7	−0.5 **	−1.5 **	−2.8 **	0.3	−3.4 **	−6.8 **	0.5	0.6	−5.5 **	−1.5 **	−1.0 **	−3.1 **
T8	−2.6 **	−0.5 **	2.0 **	−8.2 **	−1.6 *	−7.9 **	−1.6 **	−1.1 **	−0.6 *	0.2	0.3	−2.9 **
T9	−2.3 **	0.02	2.7 **	−11.8 **	−4.7 **	−4.4 **	−2.1 **	−1.9 **	−0.8 *	−0.9 **	−0.4	−1.0 **
T10	0.8 **	−0.2 **	−1.3 **	−6.5 **	−1.5 *	−0.3	4.5 **	2.3 **	2.9 **	0.02	1.1 **	1.4 **
T11	4.5 **	6.4 **	8.3 **	13.7 **	8.9 **	11.9 **	3.5 **	3.2 **	2.8 **	4.5 **	3.5 **	4.4 **
T12	1.8 **	0.2 **	−0.4 **	17.5 **	1.8 *	2.2 **	0.5	0.7 *	1.5 **	1.0 **	0.04	0.1
L.S.D. 5%	0.2	0.1	0.1	1.3	1.4	1.1	0.6	0.7	0.6	0.4	0.4	0.3
L.S.D. 1%	0.3	0.2	0.2	1.8	2.0	1.6	0.8	0.9	0.8	0.6	0.6	0.5

L.S.D.: least significant difference; **: highly significant at 1%; *: significant at 5%; 6 D: 6 days, 9 D: 9 days, 12 D: 12 days.

**Table 6 genes-13-00906-t006:** General combining ability effects of the parental lines for no. of spikelets/panicle, no. of filled grains/panicle, panicle weight and spikelet fertility percentage traits.

Traits Parents	No. of Spikelets/Panicle			No. of Filled Grains/Panicle			Panicle Weight (g)			Spikelet Fertility (%)		
	6 D	9 D	12 D	6 D	9 D	12 D	6 D	9 D	12 D	6 D	9 D	12 D
L1	24.8 **	21.3 **	16.9 **	22.4 **	18.2 **	12.1 **	0.4 **	0.32 **	0.35 **	−0.4 *	−0.7 **	−1.8 **
L2	−24.8 **	−21.3 **	−16.9 **	−22.4 **	−18.2 **	−12.1 **	−0.4 **	−0.32 **	−0.35 **	0.4 *	0.7 **	1.8 **
L.S.D. 5%	2.31	1.7	1.1	2.3	1.5	1.0	0.1	0.10	0.08	0.4	0.5	0.4
L.S.D. 1%	3.27	2.5	1.6	3.3	2.1	1.4	0.2	0.14	0.12	0.5	0.7	0.6
T1	−10.2 **	−17.9 **	−25.5 **	−24.0 **	−28.3 **	−29.6 **	−1.2 **	−1.8 **	−1.5 **	−5.9 **	−5.4 **	−4.3 **
T2	−5.5	−4.3 *	16.5 **	−0.9	2.1	19.7 **	0.0	−0.03	0.5 **	2.0 **	3.4 **	3.4 **
T3	−33.2 **	−12.1 **	11.5 **	−34.6 **	−19.8 **	4.2 **	0.1	0.2	0.8 **	−1.5 **	−4.7 **	−2.8 **
T4	−15.8 **	−15.0 **	−4.0 **	−6.3 *	−3.6 *	−2.3	0.3	0.3 *	0.02	4.3 **	6.1 **	1.2 *
T5	25.6 **	24.5 **	15.7 **	27.6 **	28.4 **	21.8 **	0.1	−0.2	−0.1	1.4 **	2.9 **	5.0 **
T6	−28.4 **	−24.3 **	−44.3 **	−23.0 **	−24.2 **	−42.9 **	−1.6 **	−0.9 **	−1.1 **	1.9 **	−0.8	−3.6 **
T7	55.5 **	34.4 **	5.9 **	58.9 **	38.8 **	10.8 **	2.9 **	1.4 **	0.6 **	2.5 **	3.3 **	4.0 **
T8	17.7 **	35.2 **	39.3 **	25.6 **	43.4 **	46.1 **	0.9 **	1.3 **	1.4 **	3.9 **	5.2 **	6.4 **
T9	34.6 **	17.2 **	15.6 **	41.8 **	23.5 **	25.3 **	1.3 **	1.2 **	0.7 **	3.9 **	4.1 **	7.3 **
T10	3.4	−16.1 **	−26.2 **	−6.7 *	−18.7 **	−27.5 **	−0.6 **	−0.5 **	−0.6 **	−4.2 **	−2.1 **	−3.3 **
T11	−44.1 **	−32.2 **	−19.7 **	−55.9 **	−45.2 **	−30.4 **	−0.1	0.2	0.1	−7.9 **	−9.7 **	−9.3 **
T12	0.3	10.4 **	15.0 **	−2.4	3.7 *	4.7 **	−1.9 **	−1.2 **	−1.0 **	−0.5	−2.3 **	−4.2 **
L.S.D. 5%	5.7	4.3	2.7	5.6	3.6	2.5	0.3	0.2	0.2	0.9	1.2	1.0
L.S.D. 1%	8.0	6.1	3.9	8.0	5.1	3.6	0.4	0.3	0.3	1.2	1.7	1.4

L.S.D.: least significant difference; **: highly significant at 1%; *: significant at 5%; 6 D: 6 days, 9 D: 9 days, 12 D: 12 days.

**Table 7 genes-13-00906-t007:** General combining ability effects of the parental lines for 1000 grain weight, grain yield/plant and grain quality traits.

Traits Parents	1000 Grain Weight (g)			Grain Yield/Plant (g)			Hulling (%)			Milling (%)			Head Rice (%)		
	6 D	9 D	12 D	6 D	9 D	12 D	6 D	9 D	12 D	6 D	9 D	12 D	6 D	9 D	12 D
L1	0.6 **	0.2 **	0.33 **	−0.6	0.1	0.4	0.13	0.07	0.01	1.7 **	1.8 **	1.3 **	4.2 **	4.5 **	4.4 **
L2	−0.6 **	−0.2 **	−0.33 **	0.6	−0.1	−0.4	−0.13	−0.07	−0.01	−1.7 **	−1.8 **	−1.3 **	−4.2 **	−4.5 **	−4.4 **
L.S.D. 5%	0.10	0.09	0.08	0.8	0.7	0.6	0.16	0.17	0.15	0.17	0.1	0.1	0.2	0.5	0.4
L.S.D. 1%	0.14	0.13	0.11	1.1	1.0	0.8	0.23	0.24	0.21	0.24	0.2	0.2	0.3	0.8	0.6
T1	−1.3 **	−1.8 **	−2.6 **	−6.8 **	−5.1 **	−8.5 **	0.1	−0.4 *	−1.2 **	−1.8 **	−2.7 **	−2.7 **	3.6 **	7.1 **	7.3 **
T2	−0.03	−0.5 **	−0.3 *	6.0 **	3.9 **	5.7 **	−0.2	−0.4 *	−0.7 **	−0.7 **	−0.5 **	−1.3 **	2.2 **	3.7 **	−2.2 **
T3	0.2	1.1 **	0.8 **	4.6 **	3.9 **	7.7 **	−1.6 **	−0.6 **	−0.5 *	−0.1	0.6 **	0.4 *	−0.5	−0.1	−0.8
T4	1.0 **	0.6 **	1.0 **	5.4 **	3.6 **	5.4 **	−0.1	−0.4	−0.2	0.003	0.2	0.8 **	0.4	0.2	10.6 **
T5	0.01	0.6 **	−0.1	−4.5 **	−4.4 **	−1.0	1.5 **	0.9 **	0.2	0.7 **	1.0 **	0.9 **	0.3	−0.1	−0.3
T6	1.1 **	0.9 **	−0.8 **	3.2 **	3.3 **	−3.0 **	−0.4 *	−0.7 **	−0.3	−0.8 **	−1.0 **	−1.6 **	−1.6 **	−0.004	0.5
T7	1.2 **	1.6 **	1.5 **	11.4 **	8.1 **	−4.7 **	−0.1	0.1	−1.3 **	−1.2 **	−1.3 **	−1.8 **	−6.1 **	−5.4 **	−12.4 **
T8	−0.1	0.7 **	1.2 **	2.5 *	3.4 **	0.3	−0.5 *	0.1	−0.4 *	−0.5 *	−0.6 **	0.3	−7.3 **	−7.7 **	−10.2 **
T9	−0.4 **	−0.2	0.9 **	2.5 *	6.4 **	14.9 **	−0.3	−0.1	1.7 **	−0.8 **	0.1	2.5 **	−8.0 **	−7.6 **	−0.6
T10	0.1	0.3 *	1.2 **	3.0 **	−0.7	−0.3	0.6 **	1.0 **	1.3 **	3.5 **	3.9 **	2.2 **	5.6 **	−0.2	−0.7
T11	0.2	−1.3 **	−1.1 **	−16.0 **	−12.5 **	−10.7 **	0.8 **	0.9 **	1.8 **	0.5 *	0.7 **	2.0 **	6.7 **	8.4 **	7.9 **
T12	−2.0 **	−2.0 **	−1.5 **	−11.3 **	−9.9 **	−5.7 **	0.4	−0.3	−0.5 *	1.3 **	−0.4 *	−1.7 **	4.8 **	1.7 *	1.2 *
L.S.D. 5%	0.2	0.2	0.2	1.9	1.8	1.4	0.4	0.4	0.4	0.4	0.3	0.3	0.6	1.3	1.1
L.S.D. 1%	0.3	0.3	0.3	2.7	2.5	2.0	0.6	0.6	0.5	0.6	0.5	0.5	0.8	1.8	1.5

L.S.D.: least significant difference; **: highly significant at 1%; *: significant at 5%; 6 D: 6 days, 9 D: 9 days, 12 D: 12 days.

**Table 8 genes-13-00906-t008:** Specific combining ability effects of the crosses for days to 50% heading, plant height, no. of panicles/plant and panicle length traits.

Traits Tester	Days to 50% Heading			Plant Height (cm)			No. of Panicles/Plant			Panicle Length (cm)		
	6 D	9 D	12 D	6 D	9 D	12 D	6 D	9 D	12 D	6 D	9 D	12 D
L1 × T1	0.15	0.09	−0.02	−3.35 **	−5.55 **	−5.12 **	6.66 **	5.95 **	4.16 **	1.14 **	0.15	0.20
L2 × T1	−0.15	−0.09	0.03	3.35 **	5.55 **	5.12 **	−6.66 **	−5.95 **	−4.16 **	−1.14 **	−0.15	−0.20
L1 × T2	−0.05	−0.36 **	−0.31 **	1.65	1.35	1.84 *	−0.45	−0.41	−0.60	−0.005	0.44	0.66 *
L2 × T2	0.05	0.36 **	0.31 **	−1.65	−1.35	−1.84 *	0.45	0.41	0.60	0.005	−0.44	−0.66 *
L1 × T3	−0.10	0.13	0.09	1.75	1.84	2.68 **	−0.87 *	−0.81	−0.20	0.22	−0.67 *	0.27
L2 × T3	0.10	−0.13	−0.09	−1.75	−1.84	−2.68 **	0.87 *	0.81	0.20	−0.22	0.67 *	−0.27
L1 × T4	−1.15 **	−1.09 **	−1.09 **	1.35	2.47 *	1.39	−0.12	−0.15	−1.00 *	−0.87 **	−0.53	−0.71 **
L2 × T4	1.15 **	1.09 **	1.09 **	−1.35	−2.47 *	−1.39	0.12	0.15	1.00 *	0.87 **	0.53	0.71 **
L1 × T5	−0.05	0.11	0.54 **	1.06	1.09	1.73 *	3.61 **	4.00 **	4.13 **	−0.40	−0.12	−0.07
L2 × T5	0.05	−0.11	−0.54 **	−1.06	−1.09	−1.73 *	−3.61 **	−4.00 **	−4.13 **	0.39	0.12	0.07
L1 × T6	0.07	0.01	−0.18 *	−8.41 **	−5.43 **	0.84	−3.64 **	−4.10 **	2.81 **	−0.45	−0.18	−0.89 **
L2 × T6	−0.07	−0.01	0.18 *	8.41 **	5.43 **	−0.84	3.64 **	4.10 **	−2.81 **	0.45	0.18	0.89 **
L1 × T7	0.05	−0.02	0.02	1.80	2.15 *	−0.26	−0.09	−0.16	−0.70	0.41	0.72 *	0.54 *
L2 × T7	−0.05	0.02	−0.02	−1.80	−2.15 *	0.26	0.09	0.16	0.70	−0.41	−0.72 *	−0.54 *
L1 × T8	0.30 *	0.16	−0.13	1.77	1.99	2.23 *	−1.09 *	−1.31 *	−1.74 **	0.45	0.61 *	0.55 *
L2 × T8	−0.30 *	−0.16	0.13	−1.77	−1.99	−2.23 *	1.09 *	1.31 *	1.74 **	−0.45	−0.61 *	−0.55 *
L1 × T9	0.25	0.03	−0.01	3.19 **	3.90 **	−0.97	−1.89 **	−2.55 **	−5.79 **	−0.09	0.01	−0.29
L2 × T9	−0.25	−0.03	0.01	−3.19 **	−3.90 **	0.97	1.89 **	2.55 **	5.79 **	0.09	−0.01	0.29
L1 × T10	−0.10	−0.17	−0.28 **	−5.81 **	−3.71 **	−3.97 **	−6.25 **	−4.10 **	−3.62 **	−0.47	−0.55	−0.37
L2 × T10	0.10	0.17	0.27 **	5.81 **	3.71 **	3.97 **	6.25 **	4.10 **	3.62 **	0.47	0.55	0.37
L1 × T11	0.45 **	0.83 **	1.01 **	2.10 *	1.47	1.17	2.48 **	1.92 **	1.00 *	−0.16	−0.10	0.17
L2 × T11	−0.45 **	−0.83 **	−1.01 **	−2.10 *	−1.47	−1.17	−2.48 **	−1.92 **	−1.00 *	0.16	0.10	−0.17
L1 × T12	0.15	0.29 **	0.34 **	2.90 **	−1.57	−1.56	1.63 **	1.70 **	1.55 **	0.23	0.20	−0.03
L2 × T12 ^‡^	−0.15	−0.29 **	−0.34 **	−2.90 **	1.57	1.56	−1.63 **	−1.70 **	−1.55 **	−0.23	−0.20	0.03
L.S.D. 5%	0.3	0.2	0.2	1.8	2.0	1.6	0.8	0.9	0.8	0.6	0.6	0.5
L.S.D. 1%	0.4	0.2	0.2	2.6	2.9	2.3	1.2	1.3	1.1	0.8	0.9	0.7

L.S.D.: least significant difference; **: highly significant at 1%; *: significant at 5%; ^‡^: control hybrid; 6 D: 6 days, 9 D: 9 days, 12 D: 12 days.

**Table 9 genes-13-00906-t009:** Specific combining ability effects of the crosses for no. of spikelets/panicle, no. of filled grains/panicle, panicle weight and spikelet fertility percentage traits.

Traits Tester	No. of Spikelets/Panicle			No. of Filled Grains/Panicle			Panicle Weight (g)			Spikelet Fertility (%)		
	6 D	9 D	12 D	6 D	9 D	12 D	6 D	9 D	12 D	6 D	9 D	12 D
L1 × T1	35.31 **	30.56 **	13.90 **	25.45 **	19.93 **	8.88 **	0.38	0.10	−0.07	−1.87 **	−3.00 **	−1.72 *
L2 × T1	−35.31 **	−30.56 **	−13.90 **	−25.45 **	−19.93 **	−8.88 **	−0.38	−0.10	0.07	1.87 **	3.00 **	1.72 *
L1 × T2	5.82	4.62	7.29 **	10.02 *	9.07 **	10.24 **	0.62 **	0.52 **	0.42 **	1.91 **	2.27 *	2.06 **
L2 × T2	−5.82	−4.62	−7.29 **	−10.02 *	−9.07 **	−10.24 **	−0.62 **	−0.52 **	−0.42 **	−1.91 **	−2.27 *	−2.06 **
L1 × T3	−21.16 **	−25.59 **	−11.21 **	−28.11 **	−35.52 **	−18.22 **	0.003	0.37 *	0.28	−4.33 **	−6.89 **	−4.67 **
L2 × T3	21.16 **	25.59 **	11.21 **	28.11 **	35.52 **	18.22 **	0.003	−0.37 *	−0.28	4.33 **	6.89 **	4.67 **
L1 × T4	−28.02 **	−24.12 **	−19.16 **	−26.71 **	−22.85 **	−29.31 **	−0.58 *	−0.59 **	−0.63 **	−0.16	−0.38	−8.08 **
L2 × T4	28.02 **	24.12 **	19.16 **	26.71 **	22.85 **	29.31 **	0.58 *	0.59 **	0.63 **	0.16	0.38	8.08 **
L1 × T5	4.63	4.19	3.74	10.54 *	9.93 **	9.83 **	1.81 **	1.44 **	1.08 **	2.43 **	2.74 **	3.54 **
L2 × T5	−4.63	−4.19	−3.74	−10.54 *	−9.93 **	−9.83 **	−1.81 **	−1.44 **	−1.08 **	−2.43 **	−2.74 **	−3.54 **
L1 × T6	−39.21 **	−24.94 **	−26.71 **	−41.10 **	−24.83 **	−25.07 **	−1.19 **	−1.05 **	−0.79 **	−2.08 **	−1.27	−2.41 **
L2 × T6	39.21 **	24.94 **	26.71 **	41.10 **	24.83 **	25.07 **	1.19 **	1.05 **	0.79 **	2.08 **	1.27	2.41 **
L1 × T7	21.89 **	16.27 **	7.89 **	27.32 **	21.15 **	9.04 **	0.52 *	0.41 *	0.33 *	2.14 **	2.45 **	0.93
L2 × T7	−21.89 **	−16.27 **	−7.89 **	−27.32 **	−21.15 **	−9.04 **	−0.52 *	−0.41 *	−0.33 *	−2.14 **	−2.45 **	−0.93
L1 × T8	16.08 **	16.28 **	15.75 **	18.99 **	19.33 **	17.73 **	0.31	0.40 *	0.43 **	1.07	1.39	1.50 *
L2 × T8	−16.08 **	−16.28 **	−15.75 **	−18.99 **	−19.33 **	−17.73 **	−0.31	−0.40 *	−0.43 **	−1.07	−1.39	−1.50 *
L1 × T9	−12.46 **	−11.33 **	−5.61 **	−12.20 **	−10.47 **	−3.22	−0.62 **	−0.57 **	0.07	−0.29	−0.07	0.80
L2 × T9	12.46 **	11.33 **	5.61 **	12.20 **	10.47 **	3.22	0.62 **	0.57 **	−0.07	0.29	0.07	−0.80
L1 × T10	−0.93	−3.16	−3.34	0.70	−0.75	2.96	0.64 **	0.52 **	0.25	1.18	1.52	4.66 **
L2 × T10	0.93	3.16	3.34	−0.70	0.75	−2.96	−0.64 **	−0.52 **	−0.25	−1.18	−1.52	−4.66 **
L1 × T11	−13.84 **	−11.80 **	−8.73 **	−11.43 **	−9.18 **	−4.67 *	−1.70 **	−1.40 **	−1.16 **	1.30 *	1.57	2.61 **
L2 × T11	13.84 **	11.80 **	8.73 **	11.43 **	9.18 **	4.67 *	1.70 **	1.40 **	1.16 **	−1.30 *	−1.57	−2.61 **
L1 × T12	31.87 **	29.02 **	26.19 **	26.52 **	24.20 **	21.83 **	−0.18	−0.14	−0.22	−1.29 *	−0.35	0.77
L2 × T12 ^‡^	−31.87 **	−29.02 **	−26.19 **	−26.52 **	−24.20 **	−21.83 **	0.18	0.14	0.22	1.29 *	0.35	−0.77
L.S.D. 5%	8.0	6.1	3.9	8.0	5.1	3.6	0.4	0.3	0.3	1.2	1.7	1.4
L.S.D. 1%	11.3	8.6	5.5	11.3	7.2	5.0	0.6	0.5	0.4	1.8	2.4	1.9

L.S.D.: least significant difference; **: highly significant at 1%; *: significant at 5%; ^‡^: control hybrid; 6 D: 6 days, 9 D: 9 days, 12 D: 12 days.

**Table 10 genes-13-00906-t010:** Specific combining ability effects of the crosses for 1000 grain weight, grain yield/plant and grain quality traits.

Traits Tester	1000 Grain Weight (g)			Grain Yield/Plant (g)			Hulling (%)			Milling (%)			Head Rice (%)		
	6 D	9 D	12 D	6 D	9 D	12 D	6 D	9 D	12 D	6 D	9 D	12 D	6 D	9 D	12 D
L1 × T1	−0.54 **	−0.24	−0.32 *	−3.88 **	−3.24 *	−2.60 *	1.57 **	1.62 **	1.39 **	1.56 **	1.30 **	1.77 **	0.70	0.13	0.38
L2 × T1	0.54 **	0.24	0.32 *	3.88 **	3.24 *	2.60 *	−1.57 **	−1.62 **	−1.39 **	−1.56 **	−1.30 **	−1.77 **	−0.70	−0.13	−0.38
L1 × T2	0.73 **	0.97 **	0.83 **	7.65 **	3.68 **	−7.10 **	0.50	0.48	0.45	−0.45	−0.45	−0.01	1.83 **	2.91 **	8.34 **
L2 × T2	−0.73 **	−0.97 **	−0.83 **	−7.65 **	−3.68 **	7.10 **	−0.50	−0.48	−0.45	0.45	0.45	0.01	−1.83 **	−2.91 **	−8.34 **
L1 × T3	0.18	0.53 **	−0.03	7.51 **	5.93 **	−1.83	−0.86 **	−0.62 *	0.62 *	1.09 **	1.47 **	0.72 **	6.27 **	6.37 **	6.51 **
L2 × T3	−0.18	−0.53 **	0.03	−7.51 **	−5.93 **	1.83	0.86 **	0.62 *	−0.62 *	−1.09 **	−1.47 **	−0.72 **	−6.27 **	−6.37 **	−6.51 **
L1 × T4	0.25	0.49 **	0.27	−0.04	2.18	0.53	−0.52	−0.64 *	−0.58 *	−1.09 **	−1.13 **	−0.15	−4.32 **	−4.61 **	−3.70 **
L2 × T4	−0.25	−0.49 **	−0.27	0.04	−2.18	−0.53	0.52	0.64 *	0.58 *	1.09 **	1.13 **	0.15	4.32 **	4.61 **	3.70 **
L1 × T5	−0.56 **	−0.49 **	−0.38 *	18.80 **	13.75 **	9.65 **	−0.42	−0.37	−0.25	0.04	0.19	0.64 *	−3.20 **	−3.64 **	−1.93 *
L2 × T5	0.56 **	0.49 **	0.38 *	−18.80 **	−13.75 **	−9.65 **	0.42	0.37	0.25	−0.04	−0.19	−0.64 *	3.20 **	3.64 **	1.93 *
L1 × T6	−0.24	−1.66 **	−2.51 **	−13.68 **	−11.33 **	1.27	−0.53	−0.71 *	−0.65 *	−0.26	−0.33	0.13	−0.26	−0.58	−0.79
L2 × T6	0.24	1.66 **	2.51 **	13.68 **	11.33 **	−1.27	0.53	0.71 *	0.65 *	0.26	0.33	−0.13	0.26	0.58	0.79
L1 × T7	−0.38 *	−1.06 **	−0.003	3.10 *	−0.93	4.33 **	0.04	0.10	0.40	−0.52	−0.59 *	−0.13	−7.03 **	−7.38 **	0.17
L2 × T7	0.38 *	1.06 **	0.003	−3.10 *	0.93	−4.33 **	−0.04	−0.10	−0.40	0.52	0.59 *	0.13	7.03 **	7.38 **	−0.17
L1 × T8	−0.39 *	−0.98 **	−0.07	−3.21 *	−0.26	−1.65	0.37	0.43	0.23	−1.18 **	−1.06 **	−0.58 *	6.04 **	5.72 **	3.98 **
L2 × T8	0.39 *	0.98 **	0.07	3.21 *	0.26	1.65	−0.37	−0.43	−0.23	1.18 **	1.06 **	0.58 *	−6.04 **	−5.72 **	−3.98 **
L1 × T9	0.83 **	0.93 **	1.04 **	−3.98 **	−0.93	2.02 *	0.34	0.76 *	−0.75 **	1.68 **	1.18 **	−3.12 **	5.89 **	5.57 **	−9.49 **
L2 × T9	−0.83 **	−0.93 **	−1.04 **	3.98 **	0.93	−2.02 *	−0.34	−0.76 *	0.75 **	−1.68 **	−1.18 **	3.12 **	−5.89 **	−5.57 **	9.49 **
L1 × T10	0.49 **	0.73 **	0.63 **	−15.57 **	−10.61 **	−6.48 **	−0.66 *	−0.74 *	−0.68 *	1.09 **	1.46 **	1.85 **	−3.62 **	−3.77 **	−2.75 **
L2 × T10	−0.49 **	−0.73 **	−0.63 **	15.57 **	10.61 **	6.48 **	0.66 *	0.74 *	0.68 *	−1.09 **	−1.47 **	−1.85 **	3.62 **	3.77 **	2.75 **
L1 × T11	−0.48 **	−0.19	−0.27	−1.13	−2.55 *	−0.39	0.27	0.31	0.38	−0.26	−0.29	0.21	0.87 *	0.56	0.76
L2 × T11	0.48 **	0.19	0.27	1.13	2.55 *	0.39	−0.27	−0.31	−0.38	0.26	0.28	−0.21	−0.87 *	−0.56	−0.76
L1 × T12	0.11	0.96 **	0.83 **	4.43 **	4.31 **	2.27 *	−0.09	−0.64 *	−0.58 *	−1.71 **	−1.77 **	−1.31 **	−3.17 **	−1.28	−1.47
L2 × T12 ^‡^	−0.11	−0.96 **	−0.83 **	−4.43 **	−4.31 **	−2.27 *	0.09	0.64 *	0.58 *	1.71 **	1.76 **	1.31 **	3.17 **	1.28	1.47
L.S.D. 5%	0.3	0.3	0.3	2.7	2.5	2.0	0.6	0.6	0.5	0.6	0.5	0.5	0.8	1.8	1.5
L.S.D. 1%	0.5	0.5	0.4	3.9	3.5	2.8	0.8	0.8	0.7	0.8	0.7	0.7	1.1	2.6	2.1

L.S.D.: least significant difference; **: highly significant at 1%; *: significant at 5%; ^‡^: control hybrid; 6 D: 6 days, 9 D: 9 days, 12 D: 12 days.

**Table 11 genes-13-00906-t011:** Percent contribution of different components (lines, testers, and lines × testers) towards the crosses’ sum of squares for various traits in rice under three irrigation regimes (6, 9, and 12 days).

Characters	Contribution of Line			Contribution of Tester			Contribution of Line × Tester		
	6 D	9 D	12 D	6 D	9 D	12 D	6 D	9 D	12 D
Days to 50% heading	4.87	5.40	2.47	91.58	91.03	95.11	3.56	3.58	2.42
Plant height (cm)	23.76	47.33	27.23	65.52	36.29	65.92	10.73	16.38	6.86
No. of panicles/plant	0.38	0.98	2.75	28.71	23.47	36.07	70.91	75.55	61.18
Panicle length (cm)	6.10	5.42	0.22	85.39	84.88	94.36	8.51	9.70	5.42
No. of spikelets/ panicle	31.96	34.18	27.65	41.22	37.75	51.54	26.82	28.06	20.81
No. of filled grains/panicle	24.73	22.75	13.63	50.00	51.08	63.16	25.27	26.16	23.21
Panicle weight (g)	6.63	6.51	10.16	62.02	57.33	60.21	31.35	36.16	29.63
Spikelet fertility (%)	0.82	1.82	7.92	78.91	74.75	62.19	20.28	23.43	29.88
1000 grain weight (g)	23.19	2.98	4.43	59.41	60.60	63.05	17.41	36.41	32.52
Grain yield/plant (g)	0.23	0.01	0.25	41.79	47.56	72.08	57.98	52.43	27.67
Hulling (%)	1.62	0.58	0.01	55.57	40.13	71.02	42.81	59.29	28.97
Milling (%)	50.07	46.76	26.76	30.16	35.96	47.02	19.76	17.27	26.22
Head rice (%)	29.75	32.46	24.09	39.27	38.11	50.68	30.98	29.43	25.23

## Data Availability

All data generated during this study is presented in the manuscript and Appendix A.

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
