# Peer review of "Combining Ability and Gene Action Controlling Agronomic Traits for Cytoplasmic Male Sterile Line, Restorer Lines, and New Hybrids for Developing of New Drought-Tolerant Rice Hybrids"

_genes, 2022, doi:10.3390/genes13050906_

Round 1
Reviewer 1 Report
I suggest to review the appropriate format for the citation in the text : eg according to X and Y (10)...
as shown by Z et al (4, 6)...
please check the spelling (so many mistakes)...

Author Response
Review Report (Reviewer 1) Attached file.

Reviewer 2 Report
In this study, the authors show interesting information that different genotypes and cross combinations displayed different agronomic performances under water deficit. The experiments are fairly designed and analyzed. However, the authors need to provide more detailed information.
- Because the authors do not mention or study on the restorer fertility genes in the manuscript, I recommended authors to modify the article title.
- In the abstract, L47, the words ‘earliness and shortness’, the authors should be explained in more detail.
- In the introduction, L107, the authors do not mention the gene action in the manuscript, I recommended authors to modify that.
- In the materials and methods, L123, the word ‘hydride’ needs to spell check
- In the materials and methods, L139-140, the sentence ‘of the spacing ……was applied’ needs to check again.
- In the results, I recommended authors to change Table S1 to Table 1. I think much important information is in Table S1. Tables 2 and 3 could be changed to the supplemental table.
- In Figure 2, I recommended authors should be improved the resolution of the figure. Why not have the standard deviation in the figure?
- In the results, L212, I recommended authors modify the word ‘harsh’ to ‘severe’.
- In the results, L213, the authors should be checked again, which one is the hybrid check variety? Is Giza178?
- In the Figure 3, it is recommended that the authors should be written in more detail. A lot of information in the figure authors did not clarify.
- In Table 4, a word ‘σ’ in the ‘genetic components’ needs to delete.
- In the results, L323, a repeated word ‘NRL11’ needs to delete.
- In the results, L517, a word ‘L’ needs to delete.
- In Table 8, the word ‘50%’ needs to add in ‘days to heading’ of the characters.
- In Figure 4, it is recommended that the authors should be written in more detail. It is not clear what information the author is trying to transfer to the reader.
Author Response
Review Report (Reviewer 2) Attached file.

Reviewer 3 Report
Thanks for your invitation to revise the manuscript “genes-1691165”. Although the manuscript provides fairly robust dataset, it is poorly written, unorganized, needs major English editing, rephrasing long sentences, and resuming uninformative sentences. The sections of Ms&Ms and results are poorly presented. I recommend revising and working with the manuscript and resubmitting again.
Specific comments:
Title
The title does not describe the text, moreover, it is written as a sentence, not as article title. It could be improved to be “Combining ability and gene action controlling agronomic traits in cytoplasmic male sterile lines under different irrigation regimes” insteat of “Utilization of the restorer lines that carry some of the major restorer fertility genes in the breeding of new drought-tolerant hybrid rice”.
Introduction
There are several grammatical mistakes incorporated throughout the manuscript. The authors should improve the sections presenting the importance of developing drought-tolerant genotypes and the role of cytoplasmic male sterile lines in this aspect. The references need to be better managed in the text following the MDPI style.
Materials and methods
This section is not well described, the first subtitle should be “Plant material and Experimental site”. More details on the Experimental site should be added. Male parents are repeated twice in Table 1, this repetition could be avoided. The used lines could be coded as L1 and L2 and applied testers T1 to T12. The complete names could be replaced by codes throughout the manuscript to be L1×T1 and L1×T12. The applied water amount should be provided for the three irrigation regimes.
Results
-The subsection of “Analysis of variance” should be presented as the first substitute of the results followed by mean performance.
-The irrigation is not presented as a studied factor in the ANOVA table (Table 2).
-The table of main performance for days to heading, plant height, and the number of panicles/plant should be presented in the main text, not as supplementary material.
- Figure 1 is empty which means the manuscript has not been revised by the authors.
- The LSD should be presented as a bar on the top of the columns of Figures 1-2.
- The section of “Heat-map analysis” (lines 268-278) should be presented at the end of the manuscript under a separate subtitle for example “Interrelationship among genotypes, irrigation regimes, and traits”. This section should be combined with the part of the principal component analysis (lines 539-544) which should be improved and extended.
- The terms K2 GCA and K2 GCA are not common to be used, please revise that term. “σσ 2 GCA” in Table 4 should be corrected to be “σ 2 GCA”
Discussion
It is unorganized and poorly written with various spelling errors such as “previoues stydies, drougth, whereth, shorteage, well be contain, detewct, conterarly”. This section is presented under one subtitle of “4.1. Effect of Water Stress on….”. It should be well revised, organized and discuss obtained results.
Author Response
Review Report (Reviewer 3) attached file.

Round 2
Reviewer 2 Report
- Why are the standard deviations in figure1 is seem the same? Please reconfirm that.
- Why are the standard deviations in figure2a is seem the same? Please reconfirm that.
- Why not have the standard deviation in figure2b?
- I recommended authors should be improved the resolution of figure 4.
- In the discussion, L640-664, the authors only describe the results from figure3 and 4, I recommended the authors move this section to ‘Results’.
Reviewer 3 Report
The bars on the top of the columns of Figures 1-2 should be clarified in the title that they re correspond to LSD (p ≤ 0.01 or 0.05).
I asked the author in my previous revision to combine the section of “Heat-map analysis” with principal component analysis. Besides, I recommend ed to present this combined part at the end of the result under a separate subtitle for example “Interrelationship among genotypes, irrigation regimes, and traits”. But the authors misunderstand that and addressed it after the discussion, although it is a part of the results.
The discussion is still under one subtitle of “4.1. Effect of Water Stress on….”. Either the authors divide the discussion into different subtitles or delete this one and leave it as a complete part without subtitles.
